

# Ecological response to collapse of the biological pump following the mass extinction at the Cretaceous-Paleogene boundary

Johan Vellekoop[1,2,*], Lineke Woelders[2,*], Sanem Açikalin[3], Jan Smit[4], Bas van de Schootbrugge[1], Ismail Ö. Yilmaz[5,6], Henk Brinkhuis[1,7], Robert P. Speijer[2]

[1] Marine Palynology, Laboratory of Palaeobotany and Palynology, Faculty of Geosciences, Utrecht University Utrecht, Utrecht, 3584 CD, The Netherlands
[2] Division of Geology, Department of Earth and Environmental Sciences, KU Leuven, Leuven-Heverlee, B-3001, Belgium
[3] Badley Ashton & Associates Ltd, Winceby House, Winceby, Horncastle, Lincolnshire, LN9 6PB, UK
[4] Department of Sedimentology and Marine geology, Faculty of Earth and Life Science, VU University Amsterdam,
Amsterdam, 1018HV, The Netherlands
[5] Department of Geological Engineering, Middle East Technical University, Ankara, Turkey
[6] Department of Geological Sciences, University of Texas at Austin, Austin, TX 78712, USA
[7] Royal Netherlands Institute for Sea Research (NIOZ), Landsdiep 4, 't Horntje, Texel, 1797 SZ, The Netherlands
*These authors contributed equally to this work

*Correspondence to*: Johan Vellekoop (johan.vellekoop@kuleuven.be)

**Abstract.** It is by now unequivocally shown that the mass extinction associated with the Cretaceous-Paleogene (K-Pg) boundary (~66 Ma) is related to the environmental effects of a large extraterrestrial impact. The biological and oceanographic consequences of the mass extinction are, however, still poorly understood. According to the Living Ocean model of D'Hondt et al. (1998), the biological crisis at the K-Pg boundary resulted in a reduction of export productivity in
the earliest Paleocene. Here, we combine organic-walled dinoflagellate cyst (dinocyst) and benthic foraminiferal analyses to provide crucial new insight into changes in the coupling of pelagic and benthic ecosystems. To this end, we perform dinocyst and benthic foraminiferal analyses on the recently discovered Tethyan K-Pg boundary section at Okçular, Northwestern Turkey, and compare the results with other K-Pg boundary sites in the Tethys. The post-impact dominance of epibenthic taxa and an increase of inferred heterotrophic dinocysts in the earliest Paleocene at Okçular are consistent with published records
from other Tethyan sites. Together, these Tethyan records indicate that during the early Paleocene more nutrients were available for the Tethyan planktonic community, whereas benthic communities were deprived of food. Hence, the post-impact phase the reduction of export productivity likely resulted in enhanced recycling of nutrients in the upper part of the water column, all along the Tethyan shelves.

**1 Introduction**

It is now commonly accepted that the Cretaceous-Paleogene (K-Pg) boundary (~66 Ma) mass-extinction was associated with the impact of a large extra-terrestrial body at Chicxulub, Yucatan, Mexico. The short- and long-term environmental



implications of this impact resulted in the extinction of a large number of biological clades (Sepkoski, 1996). Based on the fossilized remains, paleontological records suggest that approximately 50% of marine genera became extinct across the K-Pg boundary. This episode thus represents one of the largest mass-extinction events in Earth history (Sepkoski, 1996; D'Hondt, 2005). Apart from short-term global environmental consequences, such as an initial 'impact winter' phase (Alvarez et al.,

1980; Vellekoop et al., 2014; Vellekoop et al., *in press*), the event also had major long-term biological consequences. The large-scale extinctions amongst primary producers caused a major restructuring of global food webs and global carbon cycling (*D'Hondt, 2005;* Coxall et al., 2006). Moreover, a collapse in the oceanic stable carbon isotope gradient between surface and bottom waters persisted for up to a few million years (Hsu and McKenzie, 1985; Zachos et al., 1989; D'Hondt et al., 1998). Initially, the Strangelove Ocean hypothesis was invoked to explain this collapse, suggesting that primary

productivity sharply decreased or ceased immediately after the K-Pg boundary, as a consequence of the extinction of primary producers (e.g. Hsu and McKenzie, 1985).

However, modelling the carbon isotope gradient response to the extinctions suggests that productivity had to continue nearly unabated to prevent the carbon isotopic signature of the global oceans ocean from drifting toward towards that of the weathering input (Kump, 1991). In addition, both the persistency of surviving biological groups of primary producers, e.g.

dinoflagellates (e.g., Brinkhuis and Zachariasse, 1988), as well as the survival of benthic foraminifera (e.g., Culver, 2003), argue against prolonged cessation of primary productivity in the global oceans. Integration of neritic and deep-sea planktic and benthic foraminiferal carbon isotopic records suggests that the breakdown of this gradient reflects a global collapse of export productivity, i.e., the fraction of organic carbon that sinks from the photic zone to the deep ocean via the biological pump (Kump, 1991; D'Hondt et al., 1998; D'Hondt, 2005; Hain et al., 2014; Esmeray-Senlet et al., 2015), rather than the

shutdown of primary productivity. This conceptual model is generally referred to as the "Living Ocean" model (D'Hondt and Zachos, 1998; D'Hondt et al., 1998; D'Hondt, 2005). According to the Living Ocean model, total biological productivity recovered rapidly after the extinction event, but the total global export productivity from the photic zone to was reduced for hundreds of thousands of years (D'Hondt et al., 1998; D'Hondt, 2005; Birch et al., 2016).The inferred reduction in the organic flux to deep waters might be a consequence of the ecosystem reorganization that resulted from the mass extinction.

Variations in species assemblages of pelagic communities can lead to large changes in the rates of particulate export (e.g., Buesseler, 1998; Stemmann and Boss, 2012). A general reduction of the number of pelagic grazers (such as macrozooplankton) or a shift in dominance from grazers that create fecal pellets (e.g., fish) to grazers that do not produce fecal pellets, could have greatly reduced the packaging of biomass into large particles that sank to the deep ocean (D'Hondt et al., 1998; D'Hondt, 2005). Grazers that do not produce fecal pellets include, amongst others, heterotrophic dinoflagellates.

The record at Gubbio, Italy, indicates that for example in the Tethys Ocean, fish abundances fell abruptly at the K-Pg boundary, remaining depressed for millions of years (Sibert et al., 2014).

Additionally, the replacement of large Cretaceous plankic foraminiferal and calcareous nannoplankton tests by smaller early Paleocene forms (e.g., Bernaola and Monechi, 2007; Molina, 2015) might have reduced biomineral ballasting



(Armstrong et al., 2002) in the earliest Paleocene, resulting in a further reduction of the carbon flux to the ocean floor (Coxall et al., 2006).

Although numerous studies have been performed to seek evidence for the K-Pg boundary impact, the millennial-scale biotic responses to this large-scale paleoceanographic change are still poorly documented and not fully understood. Some of the most intensely studied microfossil groups used in paleoenvironmental reconstructions, such as planktonic foraminifera and calcareous nannoplankton, experienced major extinctions and subsequent radiations (e.g., Smit, 1982; Huber et al., 2002; Coxall et al., 2006; Molina, 2015; Schueth et al., 2015), hampering environmental reconstructions across the K-Pg boundary interval. In contrast, benthic foraminifera and organic-cyst producing dinoflagellates are much better suited, as they show no significant extinction above background levels at the end of the Cretaceous (Brinkhuis and Zachariasse, 1988; Culver, 2003).

A few high-resolution K-Pg boundary benthic foraminiferal and organic-walled dinoflagellate cysts (dinocyst) records have been published, particularly from the southern and western shallow margins of the Tethys Ocean, for example from Tunisia, Israel, Egypt, Spain and Morocco (Brinkhuis and Zachariasse, 1988; Eshet et al., 1992; Keller et al., 1992; Coccioni and Galeotti, 1994; Speijer and Van der Zwaan, 1996; Brinkhuis et al., 1998; Peryt et al., 2002; Alegret et al., 2003; Slimani et al., 2010; Vellekoop et al., 2015) (Figure 1). These records thus potentially provide a comprehensive, Tethyan ocean-wide portrayal of the changes in pelagic-benthic coupling across the K-Pg boundary. The benthic foraminiferal and dinocyst records from the southern margin of the Tethys do reveal indications for major, short-term oceanographic changes, including changes in for example temperature, redox and trophic conditions across the K-Pg boundary (e.g., Brinkhuis and Zachariasse, 1988; Speijer and Van der Zwaan, 1996; Brinkhuis et al., 1998). Especially quantitative benthic foraminiferal records show a strong response to the impact, generally portraying an abrupt benthic community impoverishment across the boundary. At many of these K-Pg boundary sites, after a short-lived proliferation of endobenthic forms (e.g., Coccioni and Galeotti, 1994; Alegret et al., 2015), epibenthic forms dominate the initial post-impact 'disaster' phase (Culver, 2003). Since in general endobenthic forms are considered indicative for a high flux of organic matter to the seafloor and/or relatively low oxygen conditions, and epibenthic forms indicate more oligotrophic environments (e.g., Corliss, 1985; Peryt et al., 2002; Jorissen et al., 2007; Woelders and Speijer, 2015), the post-impact abundance of epibenthic forms is often explained as food starvation at the sea floor (Culver, 2003). Following this 'disaster' phase, most benthic foraminiferal records show a relatively long recovery phase, with endobenthic forms slowly returning as diversity starts to increase again (Alegret et al., 2003; Culver, 2003), as endobenthic microhabitats supposedly diversified (Speijer and Van der Zwaan, 1996).

Although combining quantitative dinocyst and benthic foraminifera analyses could provide crucial insight into changes in pelagic-benthic coupling (e.g., Guasti et al., 2005), perhaps surprisingly, no such attempts have been made for the K-Pg boundary interval so far. Earlier studies discussed either benthic foraminiferal *or* dinoflagellate response to the K-Pg boundary, each without making an attempt to combine the results of these studies into a comprehensive, integrated ocean-wide explanation for ecological patterns observed across the boundary. In addition, although the southern margins of the Western Tethys provide a fair number of high-resolution records spanning the K-Pg boundary, no such high resolution records yet exist from the northeastern margins of the Western Tethys. To be able to provide a comprehensive, Tethys



ocean-wide portrayal of the surface and bottom water ecological changes across the K-Pg boundary and the coupling between pelagic and benthic systems, additional dinocyst and benthic foraminiferal records need to be generated from the northeastern margin of the Tethys.

The Mudurnu-Göynük Basin in the Central Sakarya Region, Turkey, provides new opportunities for high-resolution K-Pg boundary benthic foraminiferal and dinocyst records from the northern margin of the Tethys. Recently, well-preserved outcrops of ancient continental margin deposits spanning the K-Pg boundary have been discovered in this basin (Açikalin et al., 2015). These outcrops include, amongst others, the Okçular section. Here, an integration of dinocyst and benthic foraminifera records of this biostratigraphically well-constrained K-Pg boundary transition is used to provide new insights in changes in, and the relationship between, planktic and benthic communities. By linking these records to the previously generated bulk-carbonate carbon isotope record of this section (Açikalin et al., 2015) and to other benthic foraminiferal and palynological records records, the biological changes recorded will be placed in the context of the K-Pg boundary pelagic crisis and collapse of export productivity in the Tethys ocean. This integrated approach enables an evaluation of the paleoecological and paleoceanographic consequences of the early Danian 'Living Ocean' condition.

## 2 Geological setting and age assessment

The Okçular section is located in the Mudurnu-Göynük Basin (Northwestern Turkey; Figure 1). In this basin, the K-Pg boundary interval is represented by the Tarakli Formation (Saner, 1980; Altiner, 1991; Açikalin et al., 2015). In the eastern side of the basin, the upper Maastrichtian is characterized by an intercalation of mudstones and turbidites, whereas in the western side of the basin the turbidites are absent. The K-Pg boundary is marked by a reddish ejecta layer at the base of a 15-20 cm thick boundary clay layer. Throughout the basin, the lower 30-50 m of the Danian is characterized by a rhythmic alternation of fine-grained limestones and carbonate-rich mudstones (Açikalin et al., 2015). During the latest Cretaceous to earliest Paleocene this site was characterized by mixed siliciclastic-carbonate sedimentation in an outer neritic to upper bathyal environment (Açikalin et al., 2015).

The Okçular section has been analyzed for siderophile trace elements, including Ir and other platinum group elements (PGEs), bulk stable carbon isotopes, planktic foraminifera, calcareous nannofossils and dinocysts. Based on these results, a detailed biostratigraphy was defined (Açikalin et al., 2015), allowing a confident age assessment of the boundary interval. The age model shows that the section contains a chronostratigraphically complete K-Pg boundary interval. The studied interval ranges from the top part of the Maastrichtian *A. mayaroensis* Zone up to the basal part of the Danian planktic foraminiferal Zone P1b and covers globally occurring First Occurrences (FO) of dinocyst marker taxa such as *Senoniasphaera inornata*, *Damassadinium californicum* and *Carpatela cornuta*.





## 3 Materials and methods

### 3.1 Sampling

High-resolution (cm-scale) sample sets were used that were acquired during 2 field campaigns, in 2010 and in 2011. For more detail on these sampling campaigns, see Açikalin et al. (2015). The samples were split for micropaleontological and palynological analyses.

### 3.2 Foraminiferal analysis

Twenty-seven samples were processed at KU Leuven for foraminiferal studies following standard micropaleontological procedures. Of these samples, 20 samples were used for quantitative benthic foraminiferal analyses (Fig. 2). Rock samples were dried in an oven at 60°C for at least 24 hours. Depending on sample size, 4 to 60 grams of dry rock were soaked in a soda solution (50g/l $Na_2SO_4$). After disintegration, samples were washed over 2 mm and 63-μm sieves. If necessary, the tenside Rewoquat was used to clean the residues and the procedure was repeated. Clean residues were dry-sieved into three fractions: 63-125 μm, 125-630 μm and >630 μm. Representative aliquots of the >125 μm fraction were obtained, containing at least 300 benthic foraminiferal specimens. Picked specimens from this size fraction were permanently stored in Plummer slides. Benthic foraminifera were identified using the taxonomy of Cushman (1946), Cushman (1951), Kellough (1965), Aubert and Berggren (1976), Berggren and Aubert (1975) and Speijer (1994).

Benthic foraminifera are commonly used as indicators for bottom water oxygenation and trophic conditions (e.g., Jorissen et al., 2007). Here, the Benthic Foraminiferal Accumulation Rate (BFAR, number of foraminifera per $cm^2$ per kyr; see Text S2 in the supporting information for details on the estimation of the BFAR) was calculated as a semi-qualitative proxy for paleoproductivity  (Jorissen et al., 2007 and references therein). In addition, the percentage of endobenthic morphotypes was calculated using the assumed microhabitat preferences inferred from benthic foraminiferal morphotype analysis (e.g., Corliss, 1985; Corliss and Chen, 1988; Alegret et al., 2003; Woelders and Speijer, 2015). In general, endobenthic forms are considered indicative for a high flux of refractory organic matter to the seafloor and/or relatively low oxygen conditions, while abundance of epibenthic forms is considered characteristic for more oligotrophic environments (Jorissen et al., 1995; Peryt, 2004; Jorissen et al., 2007). Furthermore, the bi- and triserial endobenthic forms are particularly indicative for high food supply and low oxygenation (e.g., Bernhard, 1986; Corliss and Chen, 1988; Jorissen et al., 2007). Therefore, in this study, the percentage of endobenthic forms was calculated for each sample, as well as the percentage bi-/triserial benthic taxa, to unravel food supply and oxygenation patterns.

It should be noted that assuming such an analogue with modern fauna has limitations and shortcomings. For instance, calculating the percentage endobenthics based on morphotypes and using bi-/triserials as an indicator for hypoxia and high food supply can have exceptions (Buzas et al., 1993; Jorissen et al., 2007 and references therein).

### 3.3 Palynological analysis




In this study, the palynological data from Açikalin et al. (2015) was analyzed and interpreted. In addition, two additional samples were analyzed (OK 1.5 and OK 6) to increase the resolution of the dataset, and two samples of the Açikalin et al. (2015) dataset were re-counted (OK 2.5 and OK 250; see Supplementary Information Data Set 2). Quantative slides of the 15-250 μm fraction were used. All slides are stored in the collection of the Laboratory of Palaeobotany and Palynology,

Utrecht University, The Netherlands.

To identify major changes in the dinocyst record, morphologically closely related taxa were grouped into complexes using a similar approach to Schiøler et al. (1997); Sluijs and Brinkhuis (2009) and Machalski et al. (2016). In our study, the following morphological complexes were established: (1) the *Spiniferites* complex, combining all species of *Spiniferites* and the morphologically similar genus *Achomopshaera*; (2) *Manumiella* spp., grouping all species of *Manumiella*; (3)

hexaperidinioids, lumping all other peridinioid cysts with a hexaform archeopyle; (4) other dinocysts, which includes all other dinocyst taxa and unidentifiable dinocysts (Fig. 3).

Of these different dinocyst groups, previous studies have shown that in the Tethys in particular the hexaperidinioids show strong variations across the K-Pg boundary (Brinkhuis et al., 1998; Vellekoop et al., 2015). Based on statistical correlations between palynological records and other paleo-proxies, it has been suggested that this inferred heterotrophic group

flourished best under high-nutrient conditions in the photic zone (Eshet et al., 1994; Brinkhuis et al,. 1998; Sluijs and Brinkhuis, 2009). Therefore, in this study, abundances of hexaperidinioids are considered indicative of nutrient availability in the photic zone (see Fig. S1 and Text S1 in the supporting information for a more detailed discussion on this matter).

### 3.4 Statistical analysis

To assess changes in diversity of benthic foraminifera and dinocysts across the studied interval, the Shannon diversity index (H), the species richness per sample (S), number of specimens observed per sample (N) and the Berger-Parker index were calculated for both biological groups, following Hayek and Buzas (2013) (Fig. 4). In addition, to recognize the main faunal associations within the benthic foraminiferal data, a cluster analysis was performed, using Paired Group (UPGMA) correlation distance.

Q-mode Non Metric – Multi Dimensional Scaling (NM-MDS) was performed on the benthic foraminiferal sample compositions to assess patterns in assemblage response to K-Pg boundary perturbations. Since taxa may not have a linear response to environmental changes across the K-Pg boundary, Q-mode NM-MDS is preferred over PCA and CCA (Ramette, 2007).

## 4. Results

### 4.1 Benthic Foraminifera

The benthic foraminiferal record of the Okçular section is characterized by a major turnover across the K-Pg boundary (Figs. 2,4,5,6). Of the common taxa, 8 out of 30 (~27%) disappear across the K-Pg boundary. After the K-Pg boundary crisis, 3



new taxa appear. The estimated BFAR shows a major decrease across the boundary (Fig. 5). For the foraminiferal counts and illustrations of common forms, see Dataset S1 and Figs. S2, S3.

The cluster analysis allows the identification of 4 main clusters of benthic foraminiferal taxa, Clusters A to D (Fig. 2). Cluster C is relatively large and can be subdivided into 5 sub-clusters. Based on the succession of benthic faunal assemblages, characterized by strong changes in the Shannon diversity index (H), 4 intervals can be recognized in the benthic foraminiferal record of the Okçular section (Figs. 4, 6).

The first interval comprises the uppermost Maastrichtian and is characterized by a relatively high diversity, dominated by taxa of Cluster D. In this assemblage, the bi-/triserial benthic taxa are relatively abundant (22-28%), with characteristic taxa such as *Bulimina arkadelphiana*, *Eouvigerina subsculptura* and *Praebulimina reussi*. About 50-60% of the assemblage consists of inferred endobenthic taxa. The K-Pg boundary marks an abrupt benthic community impoverishment and the decimation of taxa of Cluster D. Above the boundary, the bi-/triserial benthic taxa virtually disappear from the record (mostly below <0.5%).

The second interval, characterized by low diversity, comprises the lowermost Danian, approximately correlative to planktic foraminiferal Zone P0. In this interval epibenthic forms are most abundant (70-90%). It is dominated by taxa of Cluster A, encompassing successive peak occurrences of the taxa *Anomalinoides praeacutus*, *Trochammina* spp. and *Cibicidoides pseudoacutus*.

The third interval covers the part of the succession approximately correlative to planktic foraminiferal Zone Pα. This interval is characterized by a recovery of diversity, although the diversity is still lower than that of the top Maastrichtian assemblage. The abundance of taxa of Cluster A slowly decreases and taxa of Cluster B, mainly represented by *Osangularia plummerae* and *Cibicidoides* sp., become abundant. Endobenthic forms recover and make up 30-40% of the total assemblage.

The fourth interval starts in the interval correlative to Zone P1a. Here, the benthic community has recovered as the diversity has stabilized and is almost similar to pre-impact values. This interval is characterized by taxa of Cluster C, a typical Paleocene Midway-type fauna (Berggren and Aubert, 1975), with representatives such as *Anomalinoides praeacutus, Coryphostoma midwayensis, Cibicidoides alleni* and *Osangularia plummerae*.

## 4.2 Palynology

Palynological samples from the Okçular site yield an abundance of palynomorphs, dominated by dinocysts and with minor contributions of acritarchs, prasinophytes, organic foraminiferal linings and terrestrial palynomorphs (Dataset S2). The dinocyst associations of the Mudurnu-Göynük Basin are relatively diverse, including components characteristic for both the Tethyan and Boreal realms (Açikalin et al., 2015). As expected, the dinocyst record does not show major changes in diversity across the K-Pg boundary (Fig. 4). There is a steady decrease in dinocyst diversity from planktic foraminiferal Zone P1a upwards, but this is probably a long-term change not related to the impact. Also the estimated Dinocyst





Accumulation Rate (DAR, number of preserved cysts per cm$^2$ per kyr; see Text S2 in the supporting information for details on the estimation of the DAR) shows no major changes across the K-Pg boundary.

Throughout the palynological record, the *Spiniferites* complex is consistently the most dominant morphogroup, in general comprising 40-50% of the total assemblage (Fig. 3). Similar to other K-Pg boundary sites worldwide (Habib and Saeedi, 2007), *Manumiella* spp. shows an episode of higher relative abundances near the top of the Maastrichtian, some 30-40 cm below the K-Pg boundary. Hexaperidinioids generally make up a relatively small component of the Maastrichtian assemblage (3-17%), but show a strong increase across the K-Pg boundary. In the boundary clay layer, correlative to planktic foraminiferal Zone P0, this group increases up to 35% of the assemblage. In the peak intervals, the hexaperidinioids are mostly represented by representatives of the genera *Senegalinium* and *Cerodinium*. After an initial drop in relative abundance in the upper half of Zone P0, this group reaches a second maximum (~35%) at the base of Zone Pα and remain relatively abundant (15-30%) up to Zone P1a. After this, the hexaperidinioids slowly decrease to the top of the studied interval. The 'other dinocysts' group, with representatives such as *Areoligera* spp., *Impagidinium* spp., *Hystrichosphaeridium tubiferum*, *Operculodinium centrocarpum* and *Palynodinium grallator*, generally makes up 25-50% of the assemblage.

## 5. Discussion

### 5.1 Tethyan benthic foraminiferal turnover sequence

The major turnover in the benthic community at Okçular is largely comparable with earlier published benthic foraminiferal records from the margins of the Tethys (Speijer and Van der Zwaan, 1996; Alegret et al., 2003; Culver, 2003 and references therein). Although the specific taxa making up the foraminiferal assemblages differ per site, other Tethyan K-Pg boundary sites with faunas from middle neritic to upper bathyal depths generally portray a similar succession of assemblages (Speijer and Van der Zwaan, 1996; Peryt et al., 2002; Culver, 2003), involving the successive occurrences of (1) a typical high diversity assemblage in the Maastrichtian; (2) a low diversity 'disaster' assemblage directly after the K-Pg boundary impact, characterized by a short-lived bloom of endobenthic taxa observed in several high-resolution records (e.g., Coccioni and Galeotti, 1994; Speijer and Van der Zwaan, 1996), followed by a dominance of epibenthic taxa; (3) a 'recovery' assemblage, characterized by an increasing diversity and returning endobenthic forms; and (4) a new, high diversity assemblage, dominated by a Paleocene, Midway-type fauna, with both epi- and endobenthic forms present.

Hence, based on this succession of benthic faunal assemblages, which are characterized by strong changes in diversity (H), the K-Pg boundary benthic foraminiferal records from the Tethys can be subdivided in four intervals (Figs. 2,4-6). These intervals I to IV approximately correspond to the uppermost Maastrichtian, planktic foraminiferal Zone P0, Zone Pα-P1a and Zone P1a-P1b, respectively, and therefore, this succession roughly follows the 5-fold sequence of Smit and Romein (1985).

The transition in benthic foraminiferal assemblages in the Tethys is also illustrated by Q-mode NM-MDS (Fig. 7). The Q-mode NM-MDS of the benthic foraminiferal sample compositions of Okçular (this study) and El Kef (Speijer and Van der



Zwaan, 1996), combined in one analysis, demonstrates that although the assemblages differ between these localities, the responses of the benthic foraminiferal assemblages to the K-Pg boundary perturbations express similar patterns across the K-Pg boundary. At both Okçular and El Kef, the benthic foraminiferal records show a rapid transition from a stable Maastrichtian assemblage to an earliest Danian disaster phase in Zone P0, followed by a gradual change back towards conditions similar to the Maastrichtian. Therefore, both the succession of benthic faunal assemblages, including diversity-indices, as well as Q-mode NM-MDS analysis provide powerful tools to allow comparison of K-Pg boundary benthic foraminiferal records from different localities.

## 5.2 Ecological responses to reduced export productivity

### 5.2.1 Okçular section

The palynological, benthic foraminiferal and bulk stable isotope records of the Okçular section show major changes across the K-Pg boundary interval (Figs. 2-7), portraying the biological crisis following the impact (D'Hondt, 2005; Esmeray-Senlet et al., 2015). Following the K-Pg boundary mass extinction, endobenthic taxa, including bi-/triserial benthic taxa, almost disappear from the benthic community and the estimated BFAR shows a major decrease (Fig. 5). This interval, approximately correlative to foraminiferal Zone P0, represents the 'disaster' phase. In the dinocyst community on the other hand, hexaperidinioids show a strong increase in relative abundance during this phase (Figs. 3, 6, 7), while the DAR remains relatively stable across the K-Pg boundary (Fig. 5).

These observed changes in the benthic foraminiferal and dinocyst communities in the Okçular record are likely caused by the major reduction of both the efficiency and strength of the biological pump, in accordance with the 'Living Ocean' model (D'Hondt and Zachos, 1998; D'Hondt et al., 1998; D'Hondt, 2005; Coxall et al., 2006). As bulk carbonate $\delta^{13}$C reflects the isotopic composition of the surface ocean, which is set by burial fractions and by the photosynthetic isotope effect (Kump, 1991; Hain et al., 2014), the excursion recorded in bulk $\delta^{13}$C records worldwide and the rapid collapse in surface to deep-ocean carbon isotope gradients likely reflect the reduction of the global intensity (i.e. efficiency) of the biological pump (Hain et al., 2014). The dramatic drop in bi-/triserial benthic taxa (Fig. 6) and major decrease in BFAR (Fig. 5) indicate that there was a coinciding drop in food supply to the benthic community, indicating that the transport of organic matter to the sea floor must have also decreased.

The decrease in benthic diversity and the blooms of opportunistic *Trochammina* spp. and *Cibicidoides pseudoacutus* suggest that the benthic community experiences additional stress during Zone P0, besides food limitation (conform Jorissen et al., 2007). This stress might include reduced oxygen levels, as was suggested for instance by Speijer and van der Zwaan (1996) who identified epibenthic *Cibicidoides pseudoacutus* as a potentially hypoxia-resistant taxon. However, no unequivocal evidence for this ecological preference of *C. pseudoacutus* was provided. Furthermore, as there is no other evidence from the investigated Okçular record pointing towards hypoxic conditions during this interval, the cause for the potential additional stress for the benthic foraminiferal community during Zone P0 remains uncertain. Organic-walled cyst producing dinoflagellates did not suffer extinctions and may have become a more important component in the earliest



Paleocene phytoplankton community. Hence, besides the reduction of the amount of organic matter transported to the sea floor, the composition of food supplied by the photic zone likely changed significantly across the K-Pg boundary as well (D'Hondt, 2005), possibly presenting an additional stress factor for the benthic community (Alegret and Thomas, 2009).

## 5.2.2 Tethys Ocean

The patterns in benthic foraminiferal and dinoflagellate response to the K-Pg boundary perturbations at Okçular appear to be characteristic for shelf section in the Tethyan Realm. In Tunisia and Spain, where both benthic foraminiferal and dinoflagellate records are available, the earliest Danian is also characterized by a decrease in endobenthic foraminifera at the sea floor (Speijer and Van der Zwaan, 1996; Peryt et al., 2002; Alegret et al., 2003) and, simultaneously, blooms of hexaperidinioids in the water column (Brinkhuis et al., 1998; Vellekoop et al., 2015; Fig. 8). In the Ouled Haddou section, Morocco, in the westernmost Tethys (Fig. 1), the lowermost Danian is also characterized by a strong increase of the hexaperidinidinoid *Senegalinium* group, up to 30% of the assemblage (Slimani et al., 2010), very similar to the Okçular record (Fig. 8).

Strikingly similar to the dinocyst record from the Okçular section, the record from El Kef, the Global Stratotype Section and Point (GSSP) of the K-Pg boundary, also shows two distinct peaks in hexaperidinioids, one in Zone P0 and one at the basal part of Zone Pα. At El Kef, the initial post-impact dominance of hexaperidinioids is nonetheless less pronounced (Fig. 8). This difference in expression might be related to small differences in paleogeographic and paleoceanographic settings between sites.

Some high resolution benthic K-Pg records in the Tethyan Realm also show a short-lived bloom of endobenthic foraminiferal taxa directly after the K-Pg boundary, sometimes accompanied by other indicators of low oxygen levels (e.g. Coccioni and Galeotti, 1994; Speijer and Van der Zwaan, 1996; Kaiho et al, 1999). This suggests that, at least locally, the sea floor was temporarily characterized by hypoxic conditions following the impact. This short-lived de-oxygenation is possibly related to the mass mortality at the K-Pg boundary, resulting in a large, short-lived flux of food to the sea floor. However, since export productivity was greatly reduced after the K-Pg boundary mass extinction, very little "new" organic matter reached the sea floor after the initial post-impact influx. Hence, after accumulated organic matter was remineralized, the benthic community starved, resulting in a transition to an epibenthic-dominated benthic fauna characteristic for well-oxygenated sea floor, similar to the post-impact fauna of the Okçular record.

The combined dinocyst records suggest that the earliest Paleocene shelves of the Tethys were characterized by an increase in nutrient availability in the photic zone, whereas coeval benthic foraminiferal records indicate a major decrease in nutrient supply to the seafloor. This inverse change in nutrient availability suggests a causal link. A reduced efficiency of the biological pump and associated decrease of the fraction of biomass transported from the photic zone to the seafloor could have resulted in high rates of nutrient recycling in the upper part of the water column (D'Hondt, 2005). This suggests that the reduction of the biological pump strength, recorded by the benthic foraminiferal record, is a consequence of the decreased efficiency of the biological pump, recorded by the carbon isotope and dinoflagellate cyst records. The strong correlation



between the bulk carbon isotope curves and the Shannon diversity index (H) of the benthic foraminiferal records at both Okçular ($R^2$=0.73, $p$<0.001) and El Kef ($R^2$=0.74, $p$<0.001) shows that there is a clear link between the changes in benthic foraminiferal assemblages and the collapse and recovery of biological pump efficiency.

As a result of the reduced biological pump efficiency, more nutrients will have been available for the earliest Paleocene phytoplankton community. At Tethyan neritic to upper bathyal sites this is indicated by the higher abundance of hexaperidinioids. Similarly, blooms of eutrophic survivor taxa of calcareous nannoplankton at open ocean sites have been suggested to be related to the build-up of nutrients in the open ocean photic zone (Schueth et al., 2015). Although the carbon isotope gradients between surface and deep waters indicate the recovery of the biological pump took hundreds of thousands of years (Zachos et al., 1989; Kump, 1991; D'Hondt et al., 1998; Coxall et al., 2006; Birch et al., 2016), our records suggest that the enhanced recycling of nutrient in the Tethys was particularly intense during two phases in the first tens of thousands years after the impact (Fig. 8).

### 5.2.3 Global responses

The decrease in export productivity following the K-Pg boundary can not only be recognized in the Tethyan Realm, but also outside this region. In for example the Atlantic Ocean, Southern Ocean and the Indian Ocean, a decrease of the biological pump strength after the K-Pg boundary was observed in neritic to abyssal environments (e.g., Thomas, 1990; Olsson et al., 1996; Hull et al., 2011; Alegret et al., 2012). As an example, the benthic foraminiferal K-Pg boundary record of Blake Nose (ODP Hole 1049C, Northwestern Atlantic; Alegret and Thomas, 2004) shows a pattern that is strikingly similar to the records in the Tethyan realm, with a strong decrease of endobenthics in Zone P0, followed by a gradual recovery across Zones Pα and P1a (Fig. 8). The similarities between the patterns in the benthic foraminiferal records of the outer neritic to upper bathyal (200-500 m) sites in the Tethys and those in the benthic foraminiferal record of the middle-lower bathyal (1500-1600 m) Blake Nose record (Alegret and Thomas, 2004) suggests that these benthic foraminiferal records record a global decrease in export productivity following the K-Pg boundary.

However, several open ocean sites, mostly in the Pacific, show an opposite trend, with several lines of evidence suggesting an increased biological pump strength at these sites (e.g., Hull et al., 2011; Alegret et al., 2012). This led Alegret et al. (2012) to conclude that the decrease in biological pump strength after the K-Pg boundary was most likely a regional instead of a global effect, arguing against the Living Ocean hypothesis. Low productivity open ocean sites like the central Pacific are characterized by entirely different ecosystem structures than more eutrophic sites, for example in the Tethyan Realm (e.g. Dortch and Packerd, 1989). Therefore, the consequence of the ecosystem reorganization resulting from the mass extinction was likely also entirely different at the open ocean Pacific (e.g., Sibert et al., 2014). Esmeray-Senlet et al. (2015) proposed the term 'Heterogeneous Ocean' for this conceptual model, as an alternative to the 'Living Ocean' model. Supposedly, the 'Heterogenous Ocean' was characterized by a strong geographic heterogeneity in the extinction patterns and food supply to the sea floor,





However, it should be noted that, although the biological pump efficiency is generally relatively high in open ocean realms such as the central Pacific, the biological pump strength, i.e. the amount of organic carbon transported from the surface to the deep, is generally very low at these sites (Honjo et al, 2008; Henson et al. 2011). Small changes in absolute biological pump strength at such a locality could therefore have had a large effect on the local, oligotrophic benthic community (Alegret and Thomas, 2009), but likely represented only a small fraction of the net amount of organic matter globally transported to the deep. Hence, even though local biological pump strength might have increased at some low productivity, deep-sea sites, the large decrease in biological pump strength recorded at many other sites means that the net amount of organic matter globally transported out of the surface ocean was likely still reduced in the post-impact world (Birch et al., 2016). Therefore, we argue that such Pacific records represent exceptions to the general pattern and that, although the global ocean response can be characterized as 'heterogeneous', the K-Pg boundary mass extinction still resulted in a reduction of the net, total amount of organic matter globally transported out of the surface ocean, while biological productivity recovered rapidly after the extinction event, in accordance with the Living Ocean hypothesis.

### 5.3 Long term recovery

Following the 'disaster' phase, the hexaperidinioid cysts at Okçular and El Kef decreased in abundance and the abundance of endobenthic benthic forms increased again, as the recovery of the benthic community was initiated. This recovery phase is approximately correlative to Zone Pα and the basal part of Zone P1a, which, according to the Paleogene age constraints of Vandenberghe et al. (2012), represents at least 300 kyrs. This duration is in agreement with the estimations provided by Birch er al. (2016). During this phase, the carbon isotope records remain well below pre-impact values and the diversity of the benthic foraminiferal community has not yet fully recovered (Figs. 4-8). This indicates that the rapid and short-lived K-Pg boundary disaster was followed by a relatively long recovery phase, in line with previous estimates of a multimillion-year biological recovery (e.g., Coxall et al., 2006). Whilst the impact-related environmental perturbations were short-lived (Kring, 2007; Vellekoop et al., 2014; Vellekoop et al., 2016), the extinctions amongst important biological groups led to a reduction of the organic flux from the photic zone to deep water, resulting in major long-term biological and paleoceanographic reorganizations. Only with the evolutionary recovery of the pelagic community governing the biological carbon pump, did export productivity start to increase again (e.g., Coxall et al., 2006; Birch et al., 2016).

### 6. Conclusions

The marine palynological, benthic foraminiferal and bulk stable isotope records of the Okçular and El Kef sections reveal major changes across the K-Pg boundary interval, portraying the biological crisis at the K-Pg boundary and subsequent recovery in the earliest Paleocene. Based on the succession of benthic faunal assemblages at a number of Tethyan shelf sites, four phases can be recognized across the K-Pg boundary interval: the Maastrichtian or pre-impact phase, a disaster phase, a recovery phase and an early Paleocene phase.



Following the K-Pg boundary impact, some localities show a large, short-lived flux of food to the sea floor, likely related to the mass mortality at the K-Pg boundary. However, since export productivity was greatly reduced after the K-Pg boundary mass extinction, it is likely that, after the initial post-impact influx, very little "new" organic matter reached the sea floor. This reduction of export productivity in the post-extinction disaster phase eventually resulted in a lower food supply to the sea floor. This presented a major stress factor for benthic organisms in the Tethys Ocean. The reduced food supply resulted in an abrupt impoverishment of benthic communities. As the downward transport of nutrients was slowed down, recycling in upper layers increased. As a result, more nutrients became available for the earliest Paleocene phytoplankton community, leading to blooms of dinoflagellates along the Tethyan shelves and blooms of calcareous nannoplankton taxa of in more open ocean sites. Our records show that the enhanced recycling of nutrients in the Tethys was particularly intense in the tens of thousands of years after the impact. Following this, the slow evolutionary recovery of the pelagic community governing the biological carbon pump resulted a gradually increasing export productivity in the hundreds of thousands of years after the impact. Hence, the integration of dinocyst and benthic foraminiferal records across the K-Pg boundary provides crucial new insights in the ecological responses to the reduction of export productivity following the mass extinction at the Cretaceous-Paleogene boundary, highlighting the direct ecological consequences of the Living Ocean conditions in the post-impact world.

*Author contributions.* Johan Vellekoop, Lineke Woelders and Henk Brinkhuis designed the research. Samples were collected in the field by Johan Vellekoop, Sanem Açikalin, Ismail Yilmaz and Jan Smit. Palynological analyses were carried out by Johan Vellekoop and Henk Brinkhuis. Benthic foraminiferal analyses were carried out by Lineke Woelders and Robert Speijer. Johan Vellekoop, Lineke Woelders and Bas van de Schootbrugge prepared the manuscript with input from all authors.

*Acknowledgements.* This work is supported by the Netherlands Organization for Scientific Research (NWO) Open Competition Grant ALWPJ/09047 to H. Brinkhuis and the Research Foundation Flanders (FWO) Grant G.0B85.13 to R. P. Speijer. We thank I. Harding, L. Kump and one anonymous reviewer for their suggestions.

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



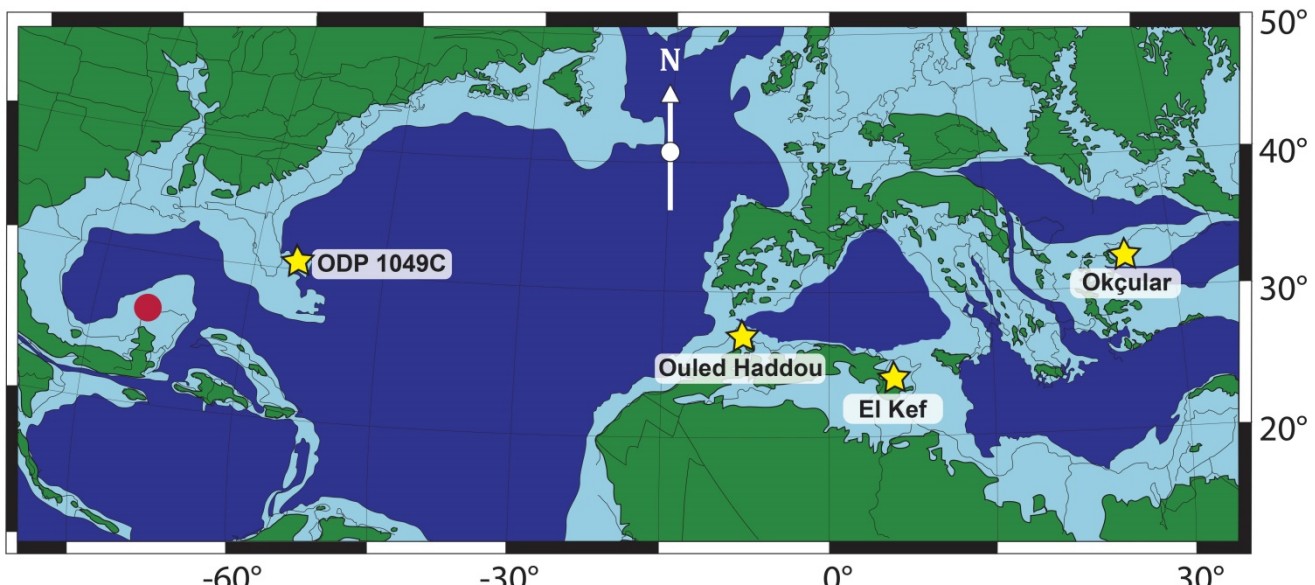

**Figure 1: A reconstruction of the late Cretaceous-early Paleogene paleogeography of the North Atlantic and Mediterranean regions, after Scotese (2004) and Scotese and Dreher (2012). The four sites that are discussed are indicated in the figure: Okçular section, Turkey (Açikalin et al., 2015; this study), El Kef, Tunisia (Brinkhuis and Zachariasse, 1988; Speijer and Van der Zwaan, 1996; Brinkhuis et al., 1998), Ouled Haddou, Morocco (Slimani et al., 2010) and ODP 1049C (Blake Nose; Alegret and Thomas, 2004).**







**Figure 2: The benthic foraminiferal record of the Okçular section. The biostratigraphy is from Açikalin et al. (2015). A cluster analysis on the benthic foraminiferal data using paired group (UPGMA) correlation distance. This cluster analysis allows the identification of 4 main clusters of benthic foraminiferal taxa, clusters A to D.**





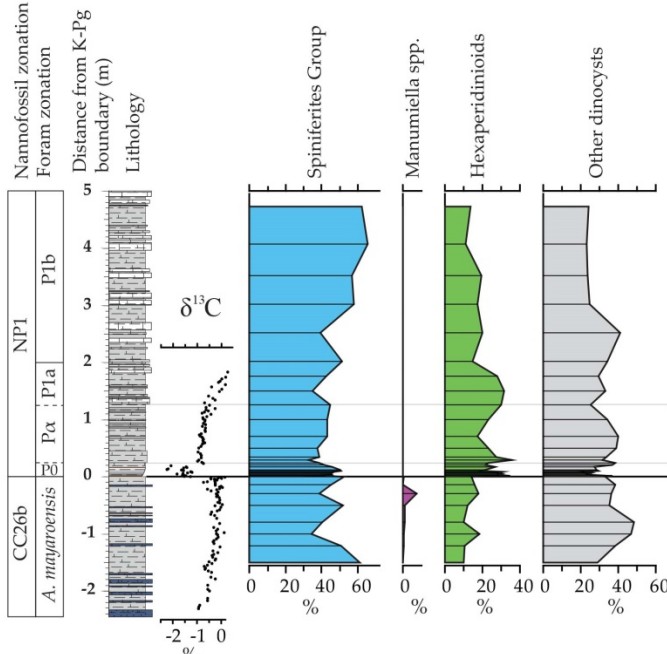

**Figure 3: The organic-walled dinoflagellate cyst record of the Okçular section. The biostratigraphy and bulk carbonate stable carbon isotope record are from Açikalin et al. (2015). The 4 main dinocyst complexes are indicated.**



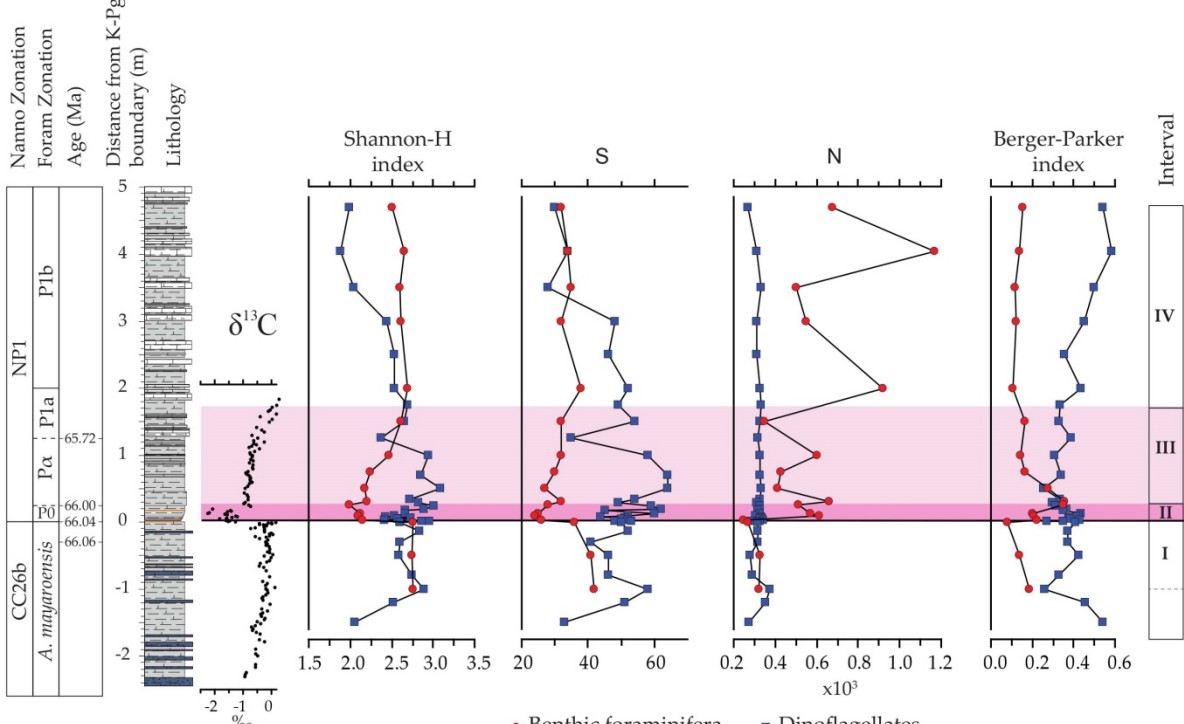

**Figure 4: Diversity indices on the benthic foraminiferal and dinocyst records of the Okçular section, with the Shannon-H diversity index, the species richness per sample (S), number of specimen observed per sample (N) and the Berger-Parker Index. As the bulk stable carbon isotope record and the Shannon-H index of the benthic foraminiferal record show a similar trend, the combination of these records can be used to subdivide the K-Pg boundary transition into 4 intervals: I, the Maastrichtian, i.e. 'pre-impact' interval; II, the direct post-impact interval, representing the 'disaster' phase; III, a 'recovery' interval and IV, the Paleocene 'post-recovery' interval. The division criteria are explained in the text.**





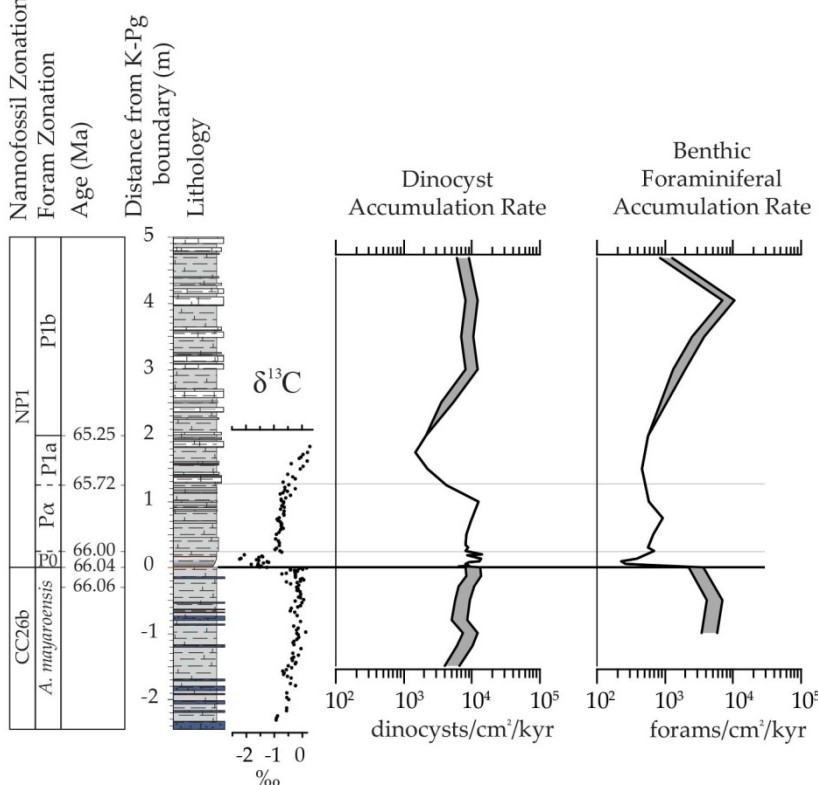

**Figure 5: Estimated Benthic Foraminiferal Accumulation Rate (BFAR; number of foraminifera per cm$^2$ sea floor per kyr) and estimated Dinocyst Accumulation Rate (DAR; number of preserved cysts per cm$^2$ sea floor per kyr) of the Okçular section. Uncertainty in estimated accumulation rates, resulting from uncertainties in sedimentation rates, is indicated in grey, providing a**
5  **range of estimated accumulation rates. For more detailed info on the calculation of the BFAR and DAR, see Text S2 of the supplementary online material.**





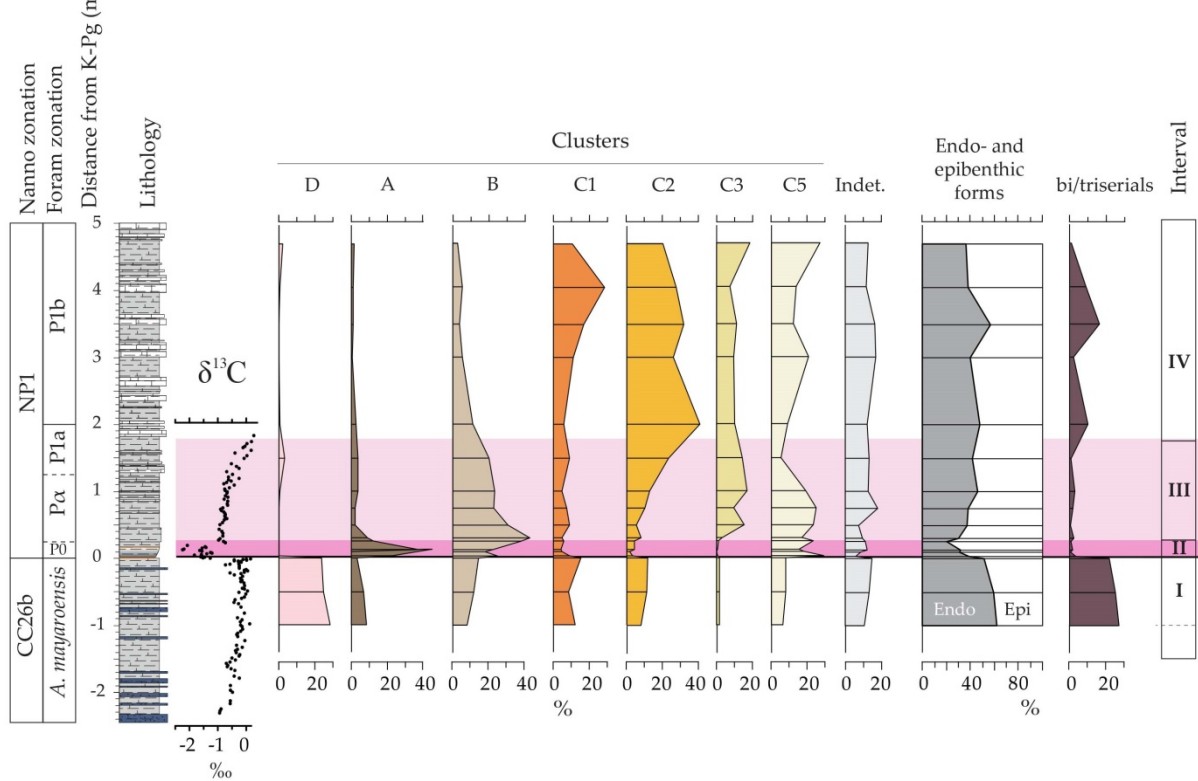

**Figure 6: Relative abundances of the 4 main clusters of benthic foraminiferal taxa, relative abundances of epibenthic and endobenthic forms and relative abundance of bi-/triserial benthic taxa.**





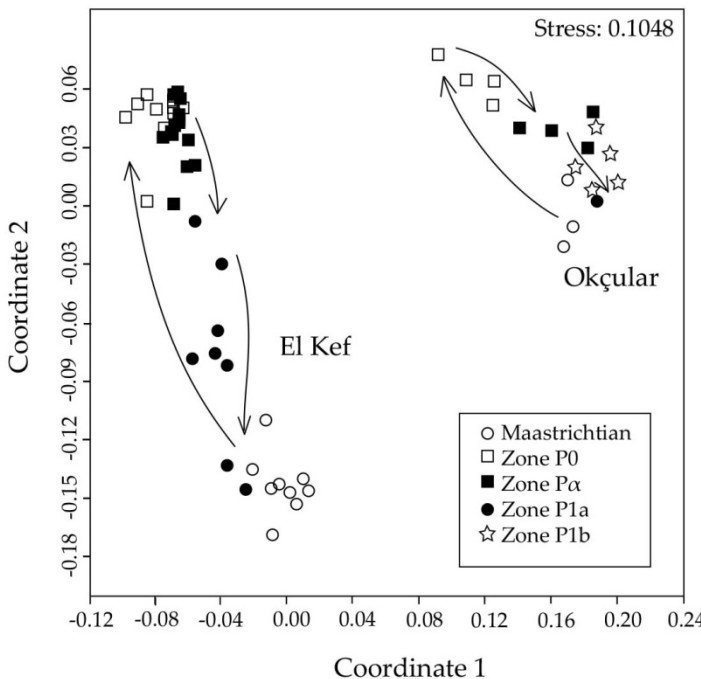

**Figure 7: Q-mode Non-Metric Multi Dimensional Scaling (on all samples in the benthic foraminiferal dataset) of Okcular (this study) and El Kef (Speijer and van der Zwaan, 1996). After an abrupt transition from latest Maastrichtian pre-impact assemblage to disaster assemblage in the Zone P0 and recovery assemblage in Zone Pα, a gradual transition towards a new equilibrium assemblage can be observed in Zones P1a and P1b.**





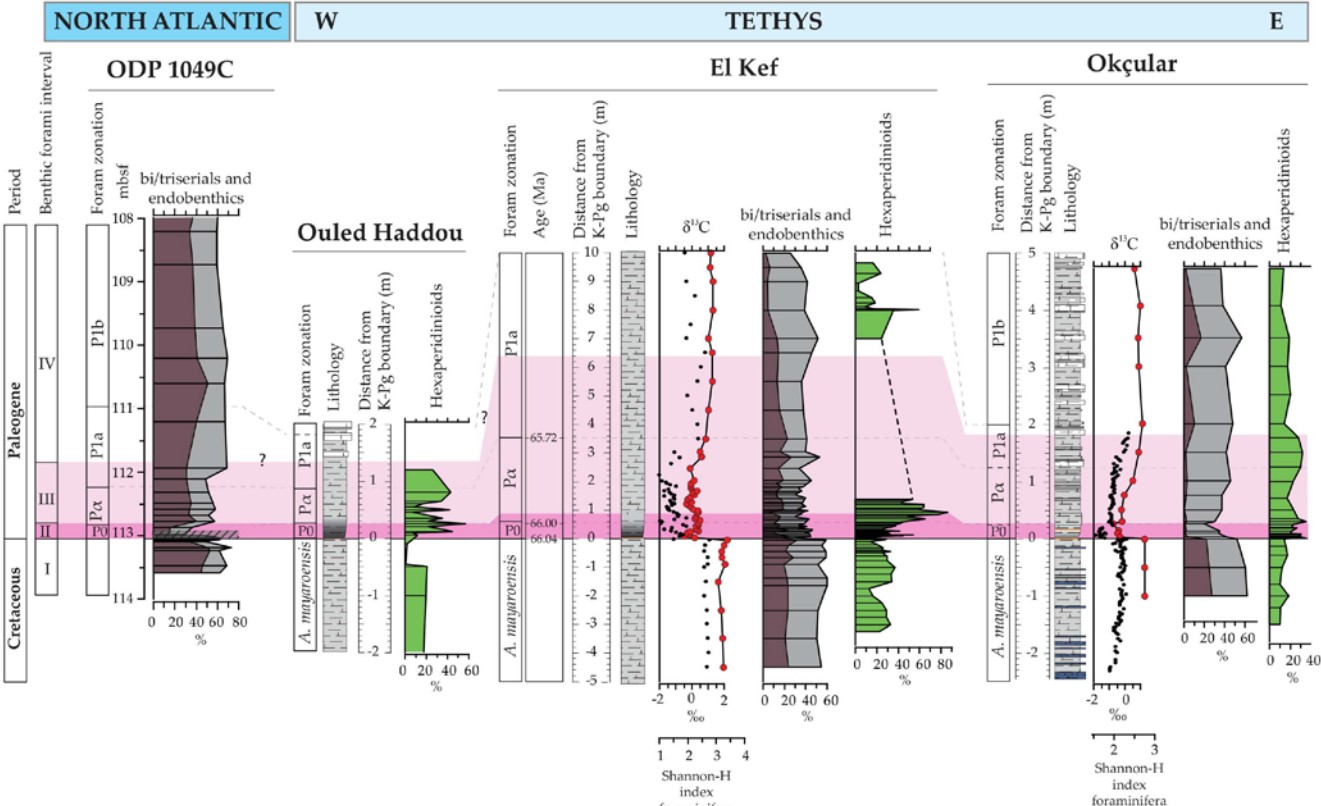

**Figure 8: The benthic foraminiferal, dinocyst and bulk stable carbon isotope records of Okçular (Turkey) and El Kef (Tunisia), the dinocyst record of Ouled Haddou (Morocco) and the benthic foraminiferal record of ODP 1049C Blake Nose (North Atlantic). Note that the definition of 'hexaperidinioids' used is provided in the text. The 4 phases identified based on the bulk stable carbon**

**isotopes and the Shannon-H index of the benthic foraminiferal records, are indicated (I-IV). The benthic foraminiferal record of ODP 1049C is from Alegret and Thomas (2004), who argued that the lowermost Paleocene of this site comprised reworked foraminifera. This interval is indicated by a dashed bar in the figure. The biostratigraphy of Ouled Haddou is from Slimani and Toufiq (2013), whereas the dinocyst record of this site is from Slimani et al. (2010). The biostratigraphy of El Kef is from Brinkhuis et al. (1998), Speijer and van der Zwaan (1996) and Molina et al. (2006). The fine fraction bulk stable carbon isotope**

**record (black dots) of El Kef is from Keller and Lindinger (1989). The Shannon-H index (red dots) and relative abundances of bi-/triserial benthic taxa (dark purple) and all endobenthic foraminifera (light purple) are based on the foraminiferal data of Speijer and van der Zwaan (1996). The palynological data of El Kef is from Brinkhuis et al. (1998) (lower part) and Guasti et al. (2005) (upper part). The Shannon-H index of the benthic foraminiferal record (red dots), the relative contribution of bi-/triserial benthic taxa (dark purple), all endobenthic foraminifera (light purple) and hexaperidinioid dinocysts from Okçular are from this study.**

**The bulk stable carbon isotope record of Okçular (black dots) is from Açikalin et al. (2015). Mbsf=meters below sea floor.**