# Peer review of "Ecological response to collapse of the biological pump following the mass extinction at the Cretaceous-Paleogene boundary"

_Biogeosciences, 2016_

## Referee Comment (RC1) · F. Boscolo Galazzo (Referee) · 22 Aug 2016

Dear Editor, I have now read the manuscript by Vellekoop et al. "Ecological response to collapse of the biological pump following the mass extinction at the Cretaceous-Paleogene boundary". The manuscript presents the first ecological data concerning marine biota disruptions following the K/Pg mass extinction from a new, continuous section spanning the K/Pg boundary in the central-western Tethys. The manuscript presents an original combination of biotic proxies in what it couples records of benthonic and planktonic marine organisms (benthic foraminifera and dinocysts) which did not suffer extinctions at the K/Pg boundary but overcame this dramatic crisis fairly well. All these aspects combined are valuable and can provide new insights on the mecha-

nisms which allowed the survival of some groups and brought to the extinction others. To achieve this aim, however, the manuscript needs, in my opinion, substantial major revisions. My main concerns regarding this manuscript are:

Research approach: it should be kept independent from the main models (e.g. Living Ocean model, see comment below) concerning the K/Pg marine biological crisis. The already known models must not be used to interpret the data, differently the reasoning gets circular preventing any new knowledge from emerging.

Benthic foraminiferal analysis: the benthic foraminiferal dataset needs to be improved. As it is right now it can provide but little information. See detailed comments below.

Comparison with benthic foraminiferal records from other sections: it cannot be done when these records are from different size fractions (125 $\mu$m or 63 $\mu$m). Either the authors stick to the 125 $\mu$m (which I do not recommend) and compare their record with the few others available within the same size fraction, or they change into the 63 $\mu$m (which I recommend), and can then make comparisons with the other 63 $\mu$m records available (included all the records from oceanic cores). However, as general suggestion, I would say the authors should focus much more on their own original data and on what new they can add, rather than on comparisons with other records.

Main Comments:

1. Title: The model arguing for a global collapse of the biological pump following the mass extinction is controversial, and still not univocally accepted (see Thomas, 2007, Birch et al., 2016). I suggest to the authors to remove it from the title.

2. Introduction: Pag. 4-L9-13: this paragraph states the approach of this paper which in my opinion is conceptually wrong. You don't do carry out a new research to place it "in the context" of what it is already known or thought to be known, but to bring in new knowledge, improve, edit or discard what's already known.

3. Methods: The benthic foraminiferal dataset should be improved in order to provide

compelling environmental and ecological reconstructions. Suggestions to improve the dataset: - The authors studied the size fraction larger than 125 $\mu$m for the benthic foraminiferal analysis. This can lead to miss important ecological information as disaster taxa and stress tolerant opportunistic taxa which bloom during environmental stress are often smaller (e.g., Boscolo Galazzo et al., 2013; Giusberti et al., 2016). To me the use of the >63 $\mu$m size fraction would have been more appropriate for this study. See for instance Thomas (1990), Alegret et al. (2003), Alegret and Thomas (2007; 2009). The study of the smaller size fraction might for instance reveal peaks of small opportunistic infaunals, challenging the current environmental interpretation. Ideally the counts should be improved counting the whole >63 um size fraction. I understand that at this stage this would imply the re-study of the whole sample set. However, the authors should at least re-count same samples using the whole >63 um size fraction in order to check that important ecological information/patterns are not missed in the critical stratigraphic intervals with the use of the larger size fraction. These additional data should be included as a figure in the paper. - To estimate benthic foraminiferal accumulation rates (BFAR) in on-land sections can be somewhat difficult as sample dry bulk density values are difficult to measure. In this work average density values derived from literature are used. For this reason, I advise caution with the use of these BFAR data to reconstruct export productivity changes, and I recommend BFAR is not used as a key parameter to interpret benthic foraminiferal faunal changes. Besides, they calculated BFAR using the number of benthic foraminifera/gr sediment for the >63 um size fraction while their assemblage counts have been done in the >125 um size fraction. This must be changed for consistency as faunal patterns in these two size fractions can be quite different. - Benthic foraminiferal counts have been made by counting 300 specimens for each samples. This is a standard counting threshold widely used in benthic foraminiferal quantitative studies. However in this specific case I encourage the use of species-specimens plots to establish the most suitable number of specimens to count (see Thomas, 1990). The use of species-specimens plots allows to ensure that species diversity is well represented. In an outer-neritic upper bathyal site species

diversity might be higher than in the deep sea settings, where the standard average of 300 specimens is usually employed. Since the relevance of diversity changes among benthic foraminifera for this study I would perform a species-specimen plot for each of the 4 intervals recognized in order to assure species diversity is well represented.

4. Results: please mention in the text (1) the number of samples along with their stratigraphic position for each of the recognized intervals (benthic foraminifera); (2) thickness and approximate duration of each interval.

5. Discussion: -Paragraph 5.1: As highlighted by the review paper of Culver (2003) and by Thomas (2007) there is no agreement regarding the ecological meaning of the generally low-diversity benthic foraminiferal assemblages occurring just above the K/Pg boundary. Even though similar changes between the % of epifaunal and infaunal species can be recognized between different records, such % changes can have different environmental meanings in different environmental settings. So I suggest caution in drawing Tethys-wide environmental scenarios based on changes in the proportion of epifaunal-infaunal species. -Paragraph 5.2.1: In this paragraph the authors seem to use the Living Ocean model (D'Hondt and Zachos, 1998) to explain their data. In my opinion they should first provide a sound interpretation of their dataset and then, argue whether their dataset fits (or not) with the main models used to explain the K/Pg $\delta$13C shift. Further, which new contributions brings their own data to a further development/understanding of these models? In my opinion this is an aspect which is currently not sufficiently addressed in the paper. -Paragraphs 5.2.2&5.2.3: As for what concern benthic foraminifera, the work cited in these paragraphs used different size fractions for their studies (either >63 um or >125 um) so direct comparisons among datasets are so far not possible.

6. Conclusion: The conclusion paragraph should be focused on summarizing the findings of the paper. Personally I think it should be rewritten highlighting the paper's data and their meaning. This is first of all a paper which presents new ecological data from a new section spanning the K/Pg boundary, it is not a review paper.

7. Figures 2-6: please add the number of samples studied and their stratigraphic position. Figure 6: please add the duration of each interval like in the previous figures.

Minor remarks: Abstract: Pag. 1-L26: beginning of the line, please insert "in" after comma. Text: Pag. 2-L13: "toward" repeated twice. Pag. 4-L11: "records" repeated twice. Pag. 5-L22: please delete "refractory".

---

## Referee Comment (RC2) · L. Kump (Referee) · 22 Aug 2016

Vellekoop present new dinocyst and benthic foraminiferal analyses from K-Pg boundary section from northwestern Turkey, and assess changes in community composition in terms of the response of the ocean's "biological pump" to environmental change that drove this mass extinction. The results are consistent with studies from other Tethyan neritic to upper bathyal sites, indicating a continuation or even enhancement of primary production but a reduction in food supply to benthic environments in the aftermath of the extinction. These results are consistent with the "living ocean" hypothesis.

The authors do a good job differentiating between the strength and the efficiency of the biological pump. In the modern ocean, strong biological pumping (i.e., high export

production) occurs in regions of strong vertical mixing or upwelling, and these areas tend to have residual surface nutrients. This means that they have inefficient but strong biological pumps. This inefficiency is reflected in a reduced gradient between surface and deep in d13C. Oligotrophic regions of the ocean, like the subtropical gyres, have low export production but highly efficient use of nutrients, and this is reflected in a large gradient in d13C. Thus, geographically at any particular time, one would expect that export production is inversely reflected in the d13C gradient. Understanding this helps understand how the collapse of this gradient at the K-Pg boundary doesn't necessarily imply a reduction in the strength of the biological pump. A good citation for this is Hilting et al., 2008, "Variations in the oceanic vertical carbon isotope gradient and their implications for the Paleocene-Eocene biological pump", Paleoceanography 23: doi:10.1029/2007PA001458.

I do have some suggestions for revision:

* The authors don't really have any direct evidence that the biological pump collapsed at their site, because they have no benthic d13C values. If they could generate these data their story would be further substantiated.

* The collapse of the biological pump should indeed lead to enhanced nutrient recycling into the photic zone and should also expand and shoal the oxygen minimum zone. The authors might want to consider this in light of their interpretations of indicators (or lack thereof) of dysoxia at various sites.

* I think the contrasting behavior in the open ocean (deep sea), e.g., page 12, paragraph beginning line 24, can be understood by the relative resistance of the more recalcitrant organic matter that the deep sea usually gets anyway to more intense surface-ocean recycling with the collapse of the biological pump. In other words, the deep-sea benthic foraminifera continue to receive recalcitrant organic matter at barely diminished rates despite the collapse of the biological pump.

Page 1, Line 16: I think the "now unequivocally shown. . ." comment about impact as

the cause of the extinction should be removed; the comment is irrelevant to the current manuscript and might be considered by some as a "pot shot" at the volcanic origin idea. The way this same idea is put on line 31 is better "It is now commonly accepted. . ."

Page 2, Line 12: My modeling did not suggest that "productivity had to continue nearly unabated . . . (Kump, 1991). Rather it showed that burial had to continue nearly unabated. Burial could have been in shallow water or on land, where the required productivity would not impact the ocean's vertical carbon isotope gradient. Primary productivity certainly COULD have continued unabated, but export productivity had to have been diminished (unless the whole ocean became destratified and well-mixed).

Line 14: "persistence"

Line 22: remove "to" after "from the photic zone"

Page 3, Line 17: "changes in, for example, temperature . . . "

Page 4, line 14: Might be good to foreshadow the main conclusions at end of this paragraph.

Page 5, line 18: "quantitative" ?

Line 22: indicative "of"

Page 6, Line 1: data "were"

Page 12, line 22: some of the effects of the impact, like the trace-metal poisoning, could have been relatively long-lived. See for example Jiang et al. Nature Geoscience 3, 280 - 285 (2010).

---

## Author Comment (AC2) · 21 Sep 2016

We thank the reviewer for his valuable comments and suggestions for our manuscript entitled "*Ecological response to collapse of the biological pump following the mass extinction at the Cretaceous-Paleogene boundary*".  Below follows a point-by-point response to the comments by the reviewer. Comments by the reviewer are **in bold**, our reply is in normal font.
* * *
**[A good citation for this is Hilting et al., 2008, "Variations in the oceanic vertical carbon isotope gradient and their implications for the Paleocene-Eocene biological pump", Paleoceanography 23:doi:10.1029/2007PA001458..**
* * *
We will incorporate this citation in our revised manuscript, although we must admit that the argumentation in this paper is quite difficult to follow.

**[The authors don't really have any direct evidence that the biological pump collapsed at their site, because they have no benthic d13C values. If they could generate these data their story would be further substantiated. ]**
* * *
We agree with the reviewer that obtaining a benthic foraminiferal $\delta^{13}C$ record would further substantiate our otherwise indirect evidence that the efficiency of the biological pump decreased at our study sites. However, given the poor preservation of carbonate in our records, with all foraminifera showing full recrystallization, it not feasible to generate a reliable benthic foraminiferal $\delta^{13}C$ record for the Okçular section.

Nevertheless, although we not have the direct evidence that a benthic foraminiferal $\delta^{13}C$ record would have provided, our records can be regarded as indirect evidence.
Our records show an increase in nutrient availability in the surface oceans, whilst there is a decrease in food supply at the sea floor. This suggests a causal link, i.e. a reduction of the transport of organic matter from the surface ocean to the sea floor (=a reduced biological pump strength), resulting from a reduction in the efficiency of the biological pump (a smaller fraction or the organic matter produced photic zone is transported down.)

Moreover, it is generally assumed that the collapse of the biological pump at the K-Pg boundary is a consequence of the ecosystem reorganization that resulted from the mass extinction. Since these extinctions occurred on a global scale, it is to be expected that they also occurred at the Okçular section, as at all previously studied sites in the Tethys, and that, as a result, the biological pump also collapsed here.

**[The collapse of the biological pump should indeed lead to enhanced nutrient recycling into the photic zone and should also expand and shoal the oxygen**

**minimum zone. The authors might want to consider this in light of their interpretations of indicators (or lack thereof) of dysoxia at various sites.]**
* * *
We agree with the reviewer that it is likely that the enhanced nutrient recycling in the photic zone should lead to an expansion of the oxygen minimum zone. In particular, when there is a reduction of transport of organic matter out of the photic zone, and, hence, more remineralization in the photic zone itself, it is to be expected that the oxygen minimum zone will shoal. It is therefore not unimaginable that the low-diversity benthic foraminiferal assemblages in P0 in our study sites might also be influenced by lower oxygen concentrations. However, in contrast to some other sites (e.g. Coccioni and Galeotti, 1994; Kaiho et al., 1999), there is no other evidence from the sites investigated here pointing towards truly hypoxic conditions. Therefore, in our study, we cannot make any firm conclusions on this matter.

It is important to realize that minimal oxygen concentrations in oxygen minimum zones depend on a large variety of factors, and therefore, at different locations, oxygen minimum zones can have very different minimal oxygen concentrations, ranging from 0 ml $l^{-1}$ at some extreme sites, to up to barely below 5 ml $l^{-1}$. Although it is very well possible that oxygen concentrations decreased at the seafloor, this decrease would not necessarily have to result in hypoxic conditions at our study sites. It is therefore possible that such a decrease in oxygen availability was not reflected in the benthic foraminiferal assemblages.

**[I think the contrasting behavior in the open ocean (deep sea), e.g., page 12, paragraph beginning line 24, can be understood by the relative resistance of the more recalcitrant organic matter that the deep sea usually gets anyway to more intense surface ocean recycling with the collapse of the biological pump. In other words, the deep-sea benthic foraminifera continue to receive recalcitrant organic matter at barely diminished rates despite the collapse of the biological pump.]**

We thank the reviewer for this valuable suggestion. This idea might provide a very good alternative hypothesis for explaining the observed differences between the signal at most continental margin sites versus the signal at Shatsky Rise. We will include a brief section discussing this alternative hypothesis.

**[Page 1, Line 16: I think the "now unequivocally shown. . ." comment about impact as the cause of the extinction should be removed; the comment is irrelevant to the current manuscript and might be considered by some as a "pot shot" at the volcanic origin idea. The way this same idea is put on line 31 is better "It is now commonly accepted. . ."]**
* * *
We agree and will adjust this sentence in our manuscript accordingly.

**[Page 2, Line 12: My modeling did not suggest that "productivity had to continue nearly unabated . . . (Kump, 1991). Rather it showed that burial had to continue nearly unabated. Burial could have been in shallow water or on land, where the required productivity would not impact the ocean's vertical carbon isotope gradient. Primary productivity certainly COULD have continued unabated, but**

**export productivity had to have been diminished (unless the whole ocean became destratified and well-mixed).]**
* * *
We agree and will adjust this paragraph in our manuscript accordingly.
* * *
**[Line 14: "persistence"]**
* * *
OK, corrected.
* * *
**[Line 22: remove "to" after "from the photic zone"]**
* * *
OK, corrected.
* * *
**[Page 3, Line 17: "changes in, for example, temperature . . . "]**
* * *
OK, corrected.
* * *
**[Page 4, line 14: Might be good to foreshadow the main conclusions at end of this Paragraph.]**
* * *
Although we understand and appreciate the suggestion by the reviewer, we feel that foreshadowing the main conclusions at the end of the introduction would give the impression that we were biased towards the outcome of this study. We therefore do not consider this the most elegant approach.
* * *
**[Page 5, line 18: "quantitative" ?]**
* * *
We thank the reviewer for pointing out this error in our manuscript. The correct term should indeed be "quantitative".
* * *
**[Line 22: indicative "of"]**
* * *
OK, corrected.
* * *
**[Page 6, Line 1: data "were"]**
* * *
OK, corrected.
* * *
**[Page 12, line 22: some of the effects of the impact, like the trace-metal poisoning, could have been relatively long-lived. See for example Jiang et al. Nature Geoscience 3, 280 - 285 (2010).]**
* * *
The reviewer is correct that, in the worst-case scenario of Jiang et al. (2010), trace-metal poisoning could have lasted for up to 10 kyrs. However, this is assuming the worst-case scenario (i.e. assuming 100% metal solubility). When more realistic metal solubilities are assumed (1% solubility), the duration of a possible trace-metal poisoning is considerably reduced, i.e. to 1 kyr (Jiang et al., 2010, supplementary material). This, indeed, is considerably longer than for example the duration of the hypothesized impact winter, but still relatively short-lived compared to the long-term biological and paleoceanographic reorganizations that occurred after the K-Pg boundary mass-extinctions, which occurred over hundreds of thousands of years.

---

## Author Response (AR1)

Dr. Johan Vellekoop
Afdeling Geologie
Departement Aard- en Omgevingswetenschappen
KU Leuven
Belgium
Johan.vellekoop@.kuleuven.be
+32 16 377780

To: *Biogeosciences*, editorial board
Leuven, October 11ᵗʰ 2016
Dear Editor,
Herewith we like to resubmit our manuscript on the ecological responses to the collapse of the
biological pump following the K-Pg boundary impact.
Please find attached a revised version of our manuscript entitled "***Ecological response to***
***collapse of the biological pump following the mass extinction at the Cretaceous-Paleogene***
***boundary***", submitted by J. Vellekoop, L. Woelders, S. Açikalin, J. Smit, B. van de Schootbrugge,
I.Ö. Yilmaz, H. Brinkhuis and R.P. Speijer.
We thank the two reviewers for their valuable comments and suggestions, which have been very
helpful in improving our manuscript. After thorough revisions, we believe that we will convince
the reviewers of our main conclusions.
The relative minor and constructive comments by the first reviewer, Lee Kump, were almost all
incorporated in the revised manuscript. Below, we include a general reply to the comments
provided by Reviewer 1.
The second reviewer, Flavia Boscola Galazzo, raised more substantial comments, regarding the
interpretation of the benthic foraminiferal data and the comparison between different
foraminiferal records. In our revised manuscript, we have clarified several issues that clearly
raised confusion and have improved several figures, as suggested by this reviewer.
Below, we include a point-by-point reply to both reviewers. We hope that by addressing all of the
points raised by the reviewers and improving our manuscript accordingly, you are able to
reconsider this manuscript for publication in *Biogeosciences.*
Looking forward to your decision on our manuscript, with best regards,
Johan Vellekoop
Signed on behalf of all the authors

Below follows a point-by-point response to the comments by the reviewers. Comments by the reviewer are **in bold**, our reply is in normal font.

**COMMENTS BY LEE KUMP**

**[A good citation for this is Hilting et al., 2008, "Variations in the oceanic vertical carbon isotope gradient and their implications for the Paleocene-Eocene biological pump", Paleoceanography 23:doi:10.1029/2007PA001458..]**

**---**
We have now incorporated this citation in our revised manuscript.

**[The authors don't really have any direct evidence that the biological pump collapsed at their site, because they have no benthic d13C values. If they could generate these data their story would be further substantiated. ]**
* * *
We agree with the reviewer that obtaining a benthic foraminiferal $\delta^{13}C$ record would further substantiate our otherwise indirect evidence that the efficiency of the biological pump decreased at our study sites. However, given the poor preservation of carbonate in our records, with all foraminifera showing full recrystallization, it not feasible to generate a reliable benthic foraminiferal $\delta^{13}C$ record for the Okçular section.

Nevertheless, although we not have the direct evidence that a benthic foraminiferal $\delta^{13}C$ record would have provided, our records do show an increase in nutrient availability in the surface oceans, whilst there is a decrease in food supply at the sea floor. This suggests a causal link, i.e. a reduction of the transport of organic matter from the surface ocean to the sea floor (=a reduced biological pump strength), resulting from a decreased fraction or the organic matter produced photic zone is transported down.

Moreover, if it is true that the collapse of the biological pump at the K-Pg boundary is a consequence of the ecosystem reorganization that resulted from the mass extinction, it is to be expected that this also occurred at the Okçular section, since these extinctions occurred on a global scale. Therefore, we consider it very likely that the Okçular section represents a similar system response as previously studied sites in the Tethys.

**[The collapse of the biological pump should indeed lead to enhanced nutrient recycling into the photic zone and should also expand and shoal the oxygen minimum zone. The authors might want to consider this in light of their interpretations of indicators (or lack thereof) of dysoxia at various sites.]**
* * *
We agree with the reviewer that it is likely that the enhanced nutrient recycling in the photic zone should lead to an expansion of the oxygen minimum zone. In particular, when there is a reduction of transport of organic matter out of the photic zone, and, hence, more remineralization in the photic zone itself, it is to be expected that the oxygen minimum zone will shoal. It is therefore not unimaginable that the low-diversity benthic foraminiferal assemblages in P0 in our study sites might also be influenced by lower oxygen concentrations. However, in contrast to some other sites (e.g. Coccioni and Galeotti, 1994; Kaiho et al., 1999), there is no other evidence from the sites investigated here pointing towards truly hypoxic conditions. Therefore, in our study, we cannot make any firm conclusions on this matter.

It is important to realize that minimal oxygen concentrations in oxygen minimum zones depend on a large variety of factors, and therefore, at different locations, oxygen minimum zones can have very different minimal oxygen concentrations, ranging from 0 ml $l^{-1}$ at some extreme sites, to up to barely below 5 ml $l^{-1}$. Although it is very well possible that oxygen concentrations decreased at the seafloor, this decrease would not necessarily have to result in hypoxic conditions at our study sites. It is therefore possible that such a decrease in oxygen availability was not reflected in the benthic foraminiferal assemblages.

We have now included a sentence in the manuscript discussing the possible shoaling of the oxygen minimum zone as a consequence of nutrient recycling in the upper part of the water column (page 10, lines 4-6).

**[I think the contrasting behavior in the open ocean (deep sea), e.g., page 12, paragraph beginning line 24, can be understood by the relative resistance of the more recalcitrant organic matter that the deep sea usually gets anyway to more intense surface ocean recycling with the collapse of the biological pump. In other words, the deep-sea benthic foraminifera continue to receive recalcitrant organic matter at barely diminished rates despite the collapse of the biological pump.]**

We thank the reviewer for this valuable suggestion. This idea might provide a very good alternative hypothesis for explaining the observed differences between the signal at most continental margin sites versus the signal at Shatsky Rise. We will include a brief section discussing this alternative hypothesis (page 12, lines 15-19).

**[Page 1, Line 16: I think the "now unequivocally shown. . ." comment about impact as the cause of the extinction should be removed; the comment is irrelevant to the current manuscript and might be considered by some as a "pot shot" at the volcanic origin idea. The way this same idea is put on line 31 is better "It is now commonly accepted. . ."]**
* * *
We agree and have now adjusted this sentence in our manuscript accordingly.

**[Page 2, Line 12: My modeling did not suggest that "productivity had to continue nearly unabated . . . (Kump, 1991). Rather it showed that burial had to continue nearly unabated. Burial could have been in shallow water or on land, where the required productivity would not impact the ocean's vertical carbon isotope**

**gradient. Primary productivity certainly COULD have continued unabated, but export productivity had to have been diminished (unless the whole ocean became destratified and well-mixed).]**
* * *
We agree and have adjust this paragraph in our manuscript accordingly.

**[Line 14: "persistence"]**
* * *
OK, corrected.

**[Line 22: remove "to" after "from the photic zone"]**
* * *
OK, corrected.

**[Page 3, Line 17: "changes in, for example, temperature . . . "]**
* * *
OK, corrected.

**[Page 4, line 14: Might be good to foreshadow the main conclusions at end of this Paragraph.]**
* * *
Although we understand and appreciate the suggestion by the reviewer, we feel that foreshadowing the main conclusions at the end of the introduction would give the impression that we were biased towards the outcome of this study. We therefore do not consider this the most elegant approach.

**[Page 5, line 18: "quantitative" ?]**
* * *
We thank the reviewer for pointing out this error in our manuscript. The correct term should indeed be "quantitative".

**[Line 22: indicative "of"]**
* * *
OK, corrected.

**[Page 6, Line 1: data "were"]**
* * *
OK, corrected.

**[Page 12, line 22: some of the effects of the impact, like the trace-metal poisoning, could have been relatively long-lived. See for example Jiang et al. Nature Geoscience 3, 280 - 285 (2010).]**

----
The reviewer is correct that, in the worst-case scenario of Jiang et al. (2010), trace-metal
poisoning could have lasted for up to 10 kyrs. However, this is assuming the worst-case
scenario (i.e. assuming 100% metal solubility). When more realistic metal solubilities
are assumed (1% solubility), the duration of a possible trace-metal poisoning is
considerably reduced, i.e. to 1 kyr (Jiang et al., 2010, supplementary material). This,
indeed, is considerably longer than for example the duration of the hypothesized impact
winter, but still relatively short-lived compared to the long-term biological and
paleoceanographic reorganizations that occurred after the K-Pg boundary mass-
extinctions, which occurred over hundreds of thousands of years.
* * *
* * *
**COMMENTS BY FLAVIA BOSCOLA GALAZZO**
**[Research approach: it should be kept independent from the main models (e.g.**
**Living Ocean model, see comment below) concerning the K/Pg marine biological**
**crisis. The already known models must not be used to interpret the data,**
**differently the reasoning gets circular preventing any new knowledge from**
**emerging.]**
--
We disagree with the reviewer here. Naturally, the benthic foraminiferal and dinocyst
data were obtained and analyzed 'independent from the main models'. However, when
interpreting the obtained results, it is, in our opinion, crucial to view the results in the
context of known models.

It should be noted that, although in the past decades there has been a considerable
debate on the occurrence of a collapse of the biological pump following the K-Pg
boundary mass extinction (e.g. Alegret and Thomas, 2009), almost all recent studies are
in agreement that post-K/Pg export productivity was reduced *to some extent*. Current
discussions mostly involve discussions on the *severity* of the reduction of export
productivity, or on the geographical differences (i.e. the 'heterogeneity' of the oceans).
Even studies that argue for a more 'heterogenic' response of the global marine
ecosystems, such as Esmerey-Senlet et al (2015) argue that, although there was spatial
heterogeneity in the wake of the K/Pg mass extinction, "*the interbasinal $\delta^{13}C$ gradient*
*was reduced after the mass extinction, suggesting* *a reduction in global export*
*productivity*".

Moreover, while records from for example the open Pacific Ocean could, potentially, be
explained as showing no reduction of the biological pump (e.g. Alegret and Thomas,
2009; Alegret et al., 2012), there is no discussion on the occurrence of a collapse of a
biological pump in the Tethys Ocean, Atlantic Ocean, Southern Ocean and Indian Ocean (e.g.
following Thomas, 1990; Olsson et al., 1996; Hull et al., 2011; Alegret et al., 2012). We therefore do not see it as the target of our study to 'test' the living ocean model. Instead, given that it has been convincingly shown that this model is valid for the Tethys (e.g. Hull et al., 2011; Alegret et al., 2012, etc.), the main focus region of this study, the objective of this study was to investigate the potential ecological consequences of this early Danian 'living ocean' condition.

To clarify these issues, we have now included a paragraph on this matter (page 2, lines 23-31).
* * *
1. **[Title: The model arguing for a global collapse of the biological pump following the mass extinction is controversial, and still not univocally accepted (see Thomas, 2007, Birch et al., 2016). I suggest to the authors to remove it from the title.]**
   **--**

Most recent studies are in agreement that, *at most sites*, export productivity was reduced *to some extent*. Also the reference cited by the reviewer (Birch et al., 2016 "Partial collapse of the marine carbon pump after the Cretaceous-Paleogene boundary"), states "*Our results show that changes in ocean circulation and foraminiferal vital effects contribute to but cannot explain all of the observed collapse in surface to deep-ocean foraminiferal d13C gradient. We conclude that the biological pump was weakened as a consequence of marine extinctions, but less severely and for a shorter duration (maximum of 1.77 m.y.) than has previously been suggested.*"

Indeed, there have been discussions on the geographical heterogeneity of the oceanic response (Hull et al., 2011; Alegret et al., 2012). However, even studies that argue for a more 'heterogenic' response of the global marine ecosystems, such as Esmerey-Senlet et al (2015) argue that, although there was spatial heterogeneity in the wake of the K/Pg mass extinction, "*the interbasinal $\delta^{13}C$ gradient was reduced after the mass extinction, suggesting a reduction in global export productivity*" and that there is convincing evidence for a collapse of a biological pump in the Tethys Ocean, Atlantic Ocean, Southern Ocean and Indian Ocean (e.g. Hull et al., 2011) Therefore, using benthic foraminiferal and dinocyst records from these regions (not the Pacific) can be used to assess the ecological response to the collapse of the biological pump in these regions. Therefore, we argue that it is valid to use this term in our title.
* * *
2. **[Introduction: Pag. 4-L9-13: this paragraph states the approach of this paper which in my opinion is conceptually wrong. You don't do carry out a new research to place it "in the context" of what it is already known or thought to be known, but to bring in new knowledge, improve, edit or discard what's already known.]**

--

We agree with the reviewer that 'placing in the context' is not a goal on itself. Naturally,
this was not the goal of this study. Therefore, we have rewritten this section accordingly.

___________________________________________________________

**3. [Methods: The authors studied the size fraction larger than 125 μm for the**
**benthic foraminiferal analysis. This can lead to miss important ecological**
**information as disas- ter taxa and stress tolerant opportunistic taxa which**
**bloom during environmental stress are often smaller (e.g., Boscolo Galazzo et**
**al., 2013; Giusberti et al., 2016). To me the use of the >63 μm size fraction**
**would have been more appropriate for this study. See for instance Thomas**
**(1990), Alegret et al. (2003), Alegret and Thomas (2007; 2009). The study of**
**the smaller size fraction might for instance reveal peaks of small opportunistic**
**infaunals, challenging the current environmental interpretation. Ideally the**
**counts should be improved counting the whole >63 um size fraction. I**
**understand that at this stage this would imply the re-study of the whole sample**
**set. However, the authors should at least re-count same samples using the**
**whole >63 um size fraction in order to check that important ecological**
**information/patterns are not missed in the critical stratigraphic intervals with**
**the use of the larger size fraction. These additional data should be included as**
**a figure in the paper. ]**
--

We fully agree with the reviewer that studying the size fraction >63 um is generally
preferable, especially in deep-sea environments with many small taxa. However, the
preservation of the studied material from Okçular is rather poor, resulting from strong
recrystallization during deep burial of the sequence. Already in the >125 um size
fraction, accurate determination of foraminifera even to the genus level was often
difficult. This resulted in a relatively high 'Indet.'group. A study on the smaller size
fraction would provide better insight into the abundance of small taxa, yet this would go
hand in hand with an even higher indeterminable fraction, further compromising an
accurate portrayal of the benthic foraminiferal assemblage. It should be noted that the
comparative data of the similar but much better preserved material of El Kef is also
based on the >125 μm fraction.

Furthermore, we are not fully convinced that the choice of size fraction necessarily
influences the overall patterns significantly. This is exemplified by the study performed
on the PETM by Ernst et al. (2006). In this study, the authors compared the results of the
separately picked size fractions (>63 and >125 μm). Although there is indeed additional
information on smaller taxa in the 63-125 μm fraction, the overall patterns prove to be
robust. This suggests that it is useful to compare assemblage changes (not absolute
numbers!) found in our study with those in other studies, even when the size fractions
used in the studies are different.

In fact, the striking similarities between the >63μm and >125μm records from different
Tethys-margin sites (e.g. Speijer and Van der Zwaan, 1996; Peryt et al., 2002; Culver,

2003; etcetera) further confirms this. Although different size fractions are used, most Tethys-margin sites portray a similar succession of assemblages.

We have now stressed this point in our revised manuscript (page 8, line25).
* * *
**[To estimate benthic foraminiferal accumulation rates (BFAR) in on-land sections can be somewhat difficult as sample dry bulk density values are difficult to measure. In this work average density values derived from literature are used For this reason, I advise caution with the use of these BFAR data to reconstruct export productivity changes, and I recommend BFAR is not used as a key parameter to interpret benthic foraminiferal faunal changes.]**

**--**

We agree with the reviewer that estimating the BFAR in on-land sections can be difficult as sample dry bulk density values are difficult to measure. However, in our study the influence of variations in dry bulk density between different stratigraphic intervals on the estimated BFAR will be minor, as the densities of mudstones, siltstones and claystones generally all fall within a range of 2-2.5 g/cm$^3$ (e.g. Manger et al., 1963). In the Okçular section, the BFAR shows a decrease of more than an order of magnitude. This is such a large decrease, that the influence of any minor variations in dry bulk density will be negligible.

Nevertheless, to exclude the possible influence of the (small) differences in dry bulk density, we have now reflected the uncertainty in BFAR, resulting from the uncertainty in dry bulk densities, in our Figure 5, indicated by the a range of accumulation rates possible with bulk densities ranging from 2 to 2.5 g/cm$^3$.
* * *
**[Besides, they calculated BFAR using the number of benthic foraminifera/gr sediment for the >63 um size fraction while their assemblage counts have been done in the >125 um size fraction. This must be changed for consistency as faunal patterns in these two size fractions can be quite different.]**
**--**

As indicated above, the choice to use the >125 um size fraction to assess changes in the benthic foraminiferal assemblages was a practical one: in this size fraction most specimens could be identified up to species level, allowing for the most accurate assessment of the benthic foraminiferal assemblage. We agree that any analyses or measurement pertaining to diversity and changes in relative abundances should be performed on the >125 um size fraction for consistency.

However, in this study, we apply the benthic foraminiferal accumulation rate as an independent, semi-quantitative proxy for productivity and/or food supply to the sea floor (cf. Jorissen et al., 2007 and references therein). This general concept is based on
the idea that as more food is available to the benthic community, there is more
ecological 'space' and, hence, more foraminifera can live on the sea floor. This does not
apply to a specific size fraction, but to all foraminifera living on the sea floor.

Therefore, in our opinion, for a proper assessment of the 'ecological state' of the benthic
community, one should include all foraminifera, not just a specific subset.

To concede to the reviewer, we have now provided both the >63 um size fraction BFAR
and the >125 um size fraction BFAR in our Data Set S2 (supplementary materials).

**[Benthic foraminiferal counts have been made by counting 300 specimens for**
**each samples. This is a standard counting threshold widely used in benthic**
**foraminiferal quantitative studies. However in this specific case I encourage the**
**use of species-specimens plots to establish the most suitable number of specimens**
**to count (see Thomas, 1990). The use of species-specimens plots allows to ensure**
**that species diversity is well represented. In an outer-neritic upper bathyal site**
**species diversity might be higher than in the deep sea settings, where the**
**standard average of 300 specimens is usually employed. Since the relevance of**
**diversity changes among benthic foraminifera for this study I would perform a**
**species-specimen plot for each of the 4 intervals recognized in order to assure**
**species diversity is well represented.]**

**--**
The reviewer argues that the number of specimens per sample may be too low to
capture the full diversity, since in an outer neritic-upper bathyal site diversity may be
higher than in the deep-sea. First of all, we note that a selection of >250 specimens per
sample is a standard procedure, both in deep-sea and continental margin studies (e.g.
Murray, 2006). Some authors recommend a higher number, e.g. counts of 300-400
specimens (Lowe and Walker, 1997). In this study, only in two cases a slightly lower
sum than 300 was obtained (245 and 269 specimens, respectively). In all other cases,
the sum was well over 300, and often over 500 (Figure 4 in the manuscript).

In other studies on deep-sea and outer neritic benthic foraminifera across the K-Pg
boundary, for instance the study from the Bidart section (Alegret et al., 2004), the Walvis
Ridge section (Alegret et al., 2007), the Agost section (Alegret et al., 2003), and the El Kef
section (Speijer and van der Zwaan, 1996), the counted number of specimens is mostly
200-300. Compared to these studies, the Fisher alpha (which calculated from number of
taxa and total specimen count per sample) and Shannon H diversity are lower in our
study, also in the cases where the counted number of specimens is higher than 600. As
Fisher-alpha should in theory not be sensitive to number of specimens counted,
especially when a relatively high sum of >250 is reached (see also the charts in Fisher et
al., 1943), this is a strong indication that diversity in our study is indeed systematically
lower than in the other sites mentioned, instead of higher. Finally, if the number of
species encountered in each sample is plotted against the number of specimens counted
per sample (SI Fig. S2), we see no trend towards higher numbers of species with increasing sample size. We thus argue that we do not have any indication that the number of specimens in our study may be too low to capture the full diversity.

To further clarify this, we have now included an additional section in our Supplementary Information (SI Text S3 and SI Figure S2).
* * *
**[4. Results: please mention in the text (1) the number of samples along with their stratigraphic position for each of the recognized intervals (benthic foraminifera); (2) thickness and approximate duration of each interval.]**
--

We are aware of the complications that arise when benthic foraminiferal data, dinocyst data or other assemblage data are plotted in graphs that are completely black, so-called 'filled silhouettes'. In such graphs, it is impossible to determine the sample depths from the graph and to determine the number of samples used to create the graph. For this reason, we decided to use colored silhouettes with horizontal black lines at each sample depth. This way, it becomes clear immediately how many samples were studied in each interval, and at which depths these samples were taken. Adding numerical data on the number of samples studied and their stratigraphic position in the figures is therefore, in our opinion, unnecessary. However, if the reader would be interested in exact sample depths and numbers of samples used, they are referred to the SI where all information is outlined in detail.

We have now indicated the recognized intervals in our Data Sets S1 and S2, so that it becomes clear which samples belong to which interval. It should be noted that both the stratigraphic depth and biostratigraphic tie-points are indicated in the figures. This allows the reader to address both the thickness and approximate durations of each interval.
* * *
**[5. Discussion: -Paragraph 5.1: As highlighted by the review paper of Culver (2003) and by Thomas (2007) there is no agreement regarding the ecological meaning of the generally low-diversity benthic foraminiferal assemblages occurring just above the K/Pg boundary. Even though similar changes between the % of epifaunal and infaunal species can be recognized between different records, such % changes can have different environmental meanings in different environmental settings. So I suggest caution in drawing Tethys-wide environmental scenarios based on changes in the proportion of epifaunal-infaunal species. ]**

**--**

We agree with the reviewer that changes between the % of epibenthic and endobenthic taxa can have different environmental meanings in different environmental settings.

Especially when one compares shelf sites, such as in the Tethys, with deep marine, open ocean sites, such as in the central pacific, the ecological meaning of, for example, high abundances of endobenthic taxa can be different.  However, particularly for this reason, we mainly compare sites in the Tethys ocean, characterized by similar palaeodepths (upper bathyal/outer neritic), similar palaeolatitudes (25-35°) and similar sedimentologies (mixed carbonate-siliciclastic sedimentation), which, therefore, represent similar environmental settings. We therefore argue that Tethys-margin-wide environmental scenarios *can* be drawn based on the recorded signals in these records.
* * *
**[Paragraph 5.2.1: In this paragraph the authors seem to use the Living Ocean model (D'Hondt and Zachos, 1998) to explain their data. In my opinion they should first provide a sound interpretation of their dataset and then, argue whether their dataset fits (or not) with the main models used to explain the K/Pg δ13C shift. ]**

**--**
The first paragraph of section 5.2.1 describes our dataset. To address the comment of the reviewer, we have now also included a couple of sentences to provide an interpretation of our dataset, independent of the Living Ocean model. The argument whether these data fit the main model, now follows in the next section.
* * *
**[Further, which new contributions brings their own data to a further development/understanding of these models? In my opinion this is an aspect which is currently not sufficiently addressed in the paper.]**

--
We do not fully understand this comment of the reviewer. In section 5.2.2, we describe what new insights our study brings, showing that, as a result of the reduced biological pump efficiency, more nutrients became available for the earliest Paleocene phytoplankton community. Our study is the first to suggest that at Tethyan neritic to upper bathyal sites this increased nutrient availability is reflected by the higher abundance of hexaperidinioids.
* * *
**[Paragraphs 5.2.2&5.2.3: As for what concern benthic foraminifera, the work cited in these paragraphs used different size fractions for their studies (either >63 um or >125 um) so direct comparisons among datasets are so far not possible. ]**

**--**
See point 3.

Note that in our study, the comparison is drawn between the Okçular and El Kef datasets, both of which are based on the >125 μm fraction. In addition, the study by Ernst et al. (2006) has convincingly shown that, although there are minor differences between >63 μm fraction and >125 μm fraction datasets, the overall biotic patterns prove to be robust. This suggests, in contrast to the reviewer's view, that comparisons between overall assemblage changes between different studies are possible, even when different size fractions are used.

The striking similarities between the >63μm and >125μm records from different localites (e.g. Speijer and Van der Zwaan, 1996; Peryt et al., 2002; Culver, 2003; etcetera) actually confirms this. Although different size fractions are used, most Tethys-margin sites portray a similar succession of assemblages.
* * *
**[6. Conclusion: The conclusion paragraph should be focused on summarizing the findings of the paper. Personally I think it should be rewritten highlighting the paper's data and their meaning. This is first of all a paper which presents new ecological data from a new section spanning the K/Pg boundary, it is not a review paper.]**

--

We differ in opinion with the reviewer on the structure of the conclusion. Over the last decades, numerous studies have been published on K-Pg boundary benthic foraminiferal and dinocyst records, from an array of sites around the world. The most novel, exciting aspect of our study is not that we are adding an extra benthic foraminiferal and dinocyst study to this collection. Rather, it is the combination of our new ecological data with existing records in the same geographical region that provides us valuable new insights in the paleoecological and paleoceanographic consequences of the early Danian 'Living Ocean' condition in this region.
* * *
**[7. Figures 2-6: please add the number of samples studied and their stratigraphic position. ]**
--
We used colored silhouettes with horizontal black lines at each sample depth. This way, it becomes clear immediately how many samples were studied in each interval, and at which depths these samples were taken. Adding numerical data on the number of samples studied and their stratigraphic position in the figures is therefore, in our opinion, superfluous. However, if the reader would be interested in exact sample depths and numbers of samples used, they are referred to the SI where all information is outlined in detail.
* * *
**[Figure 6: please add the duration of each interval like in the previous figures.]**

OK. We will include the stratigraphic tie-points in Figure 6, similar to the previous figures.
* * *
**[Minor remarks: Abstract: Pag. 1-L26: beginning of the line, please insert "in" after comma.]**

OK
* * *
**[Text: Pag. 2-L13: "toward" repeated twice. ]**

Corrected
* * *
**[Pag. 4-L11: "records" repeated twice. ]**

Corrected
* * *
**[Pag. 5-L22: please delete "refractory".]**

OK

[revised manuscript text omitted]

**Text S3.**

Compared to the benthic foraminiferal records from the Bidart section (Alegret et al., 2004), the Walvis Ridge section (Alegret and Thomas, 2007), the Agost section (Alegret et al., 2003), and the El Kef section (Speijer and van der Zwaan, 1996), the

Fisher alpha and Shannon H diversity of the benthic foraminiferal assemblages are lower in our study, also in the cases where the counted number of specimens is higher than 600. It could be argued that the number of specimens in our study (ranging from 245 to >500 specimens) may be too low to capture the full diversity of our record. However, as Fisher-alpha should in theory not be sensitive to number of specimens counted (Fisher et al., 1943), this is a strong indication that diversity in our study is systematically lower than at these other sites.

Furthermore, when the number of species encountered in each sample is plotted against the number of specimens counted per sample (SI , we see no trend towards higher numbers of species with increasing sample size. We thus argue that we do not have any indication that the number of specimens in our study may be too low to capture the full diversity.

**Figure S1**

Relative abundances of hexaperidinioids of the Okçular section palynological samples plotted against the relative abundances of sporomorphs within the same samples. This plot indicates that the variations of the relative abundances of hexaperidinioids are likely not related to changes is freshwater input at this locality.

**Figure S2**

Number of specimens counted vs number of taxa encountered, for the 4 different stratigraphic intervals. Both within each stratigraphic interval, as well as between the different intervals, there is no clear relationship between number of specimens counted and number of taxa encountered.

**Figure S3**

Plate 1. SEM images of most common benthic foraminifera found in this study.

1 a, b, c. *Angulogavelinella avnimelechi* (Reiss). Okçular, 150 cm.

2 a, b, c. *Anomalinoides* cf. *midwayensis.* Okçular, -0.5 cm.

3 a, b, c. *Anomalinoides praeacutus* (Vasilenko). Okçular, 8-9 cm.

4 a, b, c. *Cibicidoides* cf. *hyphalus.* Okçular, 49-50 cm.

5 a, b, c. *Cibicidoides pseudoacutus* (Nakkady). Okçular, 8-9 cm.

6 a, b, c. *Cibicidoides* sp. 1. Okçular, 74-75 cm.

7 a, b, c. *Valvalabamina depressa* (Alth). Okçular, 49-50 cm.

8 a, b, c. *Pulsiphonina prima* (Plummer). Okçular, 99-100 cm.

9 a, b, c. *Gyroidinoides girardanus* (Reuss). Okçular, 350 cm.

10 a, b, c. *Osangularia plummerae* (Brotzen). Okçular, 29-30 cm.

**Figure S4**

Plate 2. SEM images of most common benthic foraminifera found in this study.

11 a, b, c. *Cibicidoides alleni* (Plummer). Okçular, -0.5 cm.

12. *Bolivinoides draco draco* (Marsson). Okçular, -50 cm.

13. *Bulimina arkadelphiana* (Cushman and Parker). Okçular, -100 cm.

14. *Bulimina srobila* (Marie). Okçular, -0.5 cm.

15. *Sitella carseyae* (Plummer). Okçular, -0.5 cm.

16. *Eouvigerina subsculptura* (McNeil and Caldwell). Okçular, -50 cm.

17. *Coryphostoma midwayensis* (Cushman). Okçular, 350 cm.

a, b. *Dorothia oxycona* (Reuss). Okçular, 470 cm.

19. *Praebulimina reussi* (Morrow). Okçular, -100 cm.

20. *Gaudryina pyramidata* (Reuss). Okçular, 49-50 cm.

21. *Oolina orbignyana* (Kellough). Okçular, 150 cm.

.

**Figure S1**

[Figure]

**Figure S2**

**Figure S3**

[Figure]

 **Figure S3**

[Figure]

**Additional Supporting Information**

Captions for Data Sets S1 and S2

**Data Set S1**

Benthic foraminiferal counts of Okçular (Excell file)

**Data Set S2**

Palynological counts of Okçular (Excell file)

---

## Author Response (AR2)

Dr. Johan Vellekoop
Afdeling Geologie
Departement Aard- en Omgevingswetenschappen
KU Leuven
Belgium
Johan.vellekoop@.kuleuven.be
+32 16 377780

To: *Biogeosciences*, editorial board

Leuven, December 7th 2016

Dear Editor,

Herewith we like to resubmit our manuscript on the ecological responses to the collapse of the biological pump following the K-Pg boundary impact.

Please find attached a revised version of our manuscript entitled "*Ecological response to collapse of the biological pump following the mass extinction at the Cretaceous-Paleogene boundary*", submitted by J. Vellekoop, L. Woelders, S. Açikalin, J. Smit, B. van de Schootbrugge, I.Ö. Yilmaz, H. Brinkhuis and R.P. Speijer.

We thank the reviewer for the valuable comments, which have been very helpful in improving our manuscript. After thorough revisions, following the suggestions of the editor, we believe that we will convince the reviewer of our main conclusions.

The main issue raised by the reviewer concerns the size fraction used for our benthic foraminiferal analyses. It is sometimes suggested that different size fractions cannot be directly compared, because the different size fractions sometimes harbor different ecological information (e.g. Giusberti et al., 2016). We have now constructed a careful argumentation to address this issue. In addition, the main comparison drawn in our manuscript now includes two records with a similar size fraction. We further have stressed that the striking similarities shown in the benthic foraminiferal data from the >125 µm and >63 µm size fractions provide compelling evidence for a strong disruption of the benthic ecosystems.

We hope that by addressing this point raised by the reviewer and improving our manuscript accordingly, you are able to reconsider this manuscript for publication in *Biogeosciences.*

Below, we include a the revised manuscript text with marked changes.

Looking forward to your decision on our manuscript, with best regards,

Johan Vellekoop
        Signed on behalf of all the authors

[revised manuscript text omitted]